# THERMEVAL: A STRUCTURED BENCHMARK FOR ZERO-SHOT EVALUATION OF VISION-LANGUAGE MODELS ON THERMAL IMAGERY

## ABSTRACT

Vision-Language Models (VLMs) achieve strong results on RGB imagery, yet their ability to reason over thermal data remains largely unexplored. Thermal imaging is critical in domains where RGB fails, such as surveillance, rescue, and medical diagnostics, but existing benchmarks do not capture its unique properties. We introduce **ThermEval-B**, a benchmark of 50,000 visual question–answer pairs for evaluating zero-shot performance of open-source VLMs on thermal imagery across tasks including modality identification, human counting, temperature reasoning, and temperature estimation. ThermEval-B integrates public datasets such as LLVIP and FLIR-ADAS with our new dataset **ThermEval-D**, the first to provide per-pixel temperature annotations across diverse environments. Our evaluation reveals that while VLMs reliably distinguish raw thermal from RGB images, their performance collapses on temperature reasoning and estimation, and modality recognition becomes unreliable under false colormap renderings. Models frequently default to language priors or fixed outputs, exhibit systematic biases, or refuse to answer when uncertain. These recurring failure modes highlight thermal reasoning as an open challenge and motivate benchmarks like ThermEval-B to drive progress beyond RGB-centric evaluation.

## 1 INTRODUCTION

Computer vision research has largely centered on RGB imagery, which captures reflected visible light with rich color and texture cues. Thermal infrared imaging, by contrast, measures emitted radiation and encodes temperature, producing representations that lack many of the cues conventional models exploit. While recent vision-language models (VLMs) achieve strong zero-shot performance on RGB benchmarks, their ability to generalize to thermal imagery remains unclear. This gap raises a central question: **Can VLMs trained predominantly on RGB data reason effectively about temperature-specific tasks in thermal imagery?** The absence of benchmarks that target thermal understanding prevents the community from addressing this question systematically.

To fill this gap, we introduce **ThermEval**, which consists of a benchmark (ThermEval-B) and a dataset (ThermEval-D) for evaluating VLMs on thermal imagery. ThermEval-B defines tasks that capture both core challenges and real-world applications, including modality identification, human counting, temperature-based reasoning, and per-pixel and semantic temperature estimation. Unlike multiple-choice formats that can be solved through textual cues, our benchmark employs classification and regression tasks that require precise predictions and reasoning grounded in visual input.

ThermEval-B comprises seven tasks with over 50,000 expert-labeled visual question–answer pairs. The tasks are organized to increase in difficulty, beginning with modality identification and colormap robustness, and progressing through human counting, colorbar localization, thermal reasoning, absolute temperature estimation, and temperature interpretation at multiple depths. Together, these tasks are designed to probe complementary aspects of thermal understanding and to ensure that models attend directly to thermal signals rather than relying on language-based heuristics.

We evaluate 21 VLMs spanning compact 4B models to those exceeding 200B parameters, covering both open source and closed source families Our results show that while VLMs can reliably distinguish raw thermal from RGB images, their performance drops substantially on tasks requiring

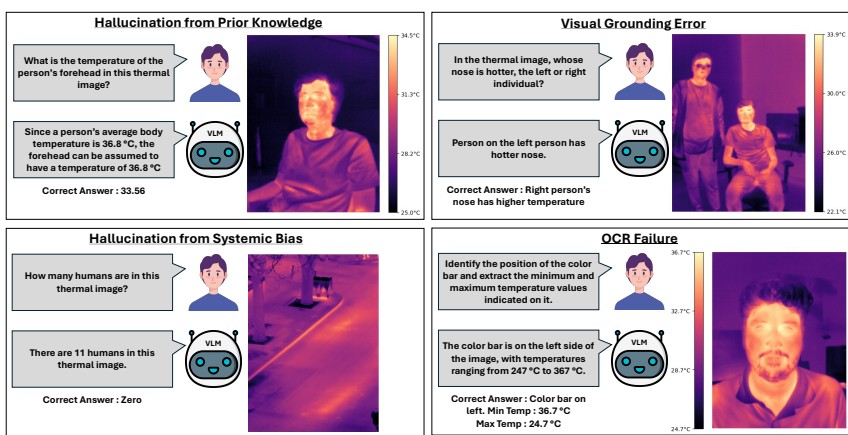

Figure 1: **Failure cases of vision-language models on thermal tasks.** Models often rely on language priors rather than thermal input, show systematic biases and hallucinations, or misinterpret the colorbar, leading to incorrect predictions Li et al. (2024).

temperature reasoning or estimation, and even modality recognition becomes unreliable under false-colormap renderings. On reasoning tasks, many models rely on language priors instead of thermal cues, producing plausible but incorrect answers, such as consistently predicting that the forehead is warmer than the nose or defaulting to 36.8°C. Others exhibit systematic biases, repeatedly outputting fixed values like 0°C, 273 K, or the number 11, regardless of the scene. A few models refuse to answer when uncertain, though this safeguard is inconsistent across architectures. These failure modes appear across model scales, indicating that the limitation lies in cross-modal grounding rather than model size. Finetuning Qwen-VL 2.5 (7B) improves performance to near-human levels on most tasks, but even after finetuning the model remains insufficiently reliable for real-world thermal applications. Overall, these findings show that thermal reasoning remains an open challenge for current VLMs and highlight the need for dedicated benchmarks that surface and diagnose such limitations.

**This work makes the following contributions:**

1. We present **ThermEval-B**, a benchmark of 50,000+ thermal VQA pairs across seven tasks split between three datasets, providing the first systematic evaluation of VLMs on thermal imagery and revealing critical gaps in temperature reasoning.
2. We introduce **ThermEval-D**, a dataset of over 500 thermal images with per-pixel temperature maps and body-part annotations across indoor and outdoor scenarios, supporting around 8.5k VQA pairs and enabling more realistic and comprehensive benchmarking than prior datasets.

## 2 RELATED WORK

Vision-language models (VLMs) have demonstrated strong performance on RGB imagery, supported by benchmarks such as MME, MMBench, SEED-Bench, and MMVet Fu et al. (2023); Liu et al. (2023); Zhang et al. (2023); Zheng et al. (2024), which evaluate perception, reasoning, and problem-solving across diverse domains. More recent benchmarks, including NaturalBench Li et al. (2024) and ZeroBench Roberts et al. (2025), further challenge VLMs with adversarial samples and complex reasoning tasks. Despite these advances, existing evaluations remain largely RGB-centric and do not assess performance on alternative sensing modalities.

We organize related work into two categories. The first covers benchmarks developed for thermal and multispectral modalities, while the second reviews available thermal and infrared datasets.

### 2.1 THERMAL AND OTHER MULTI-SPECTRAL BENCHMARKS

Multispectral modalities encode distinct physical signals: thermal reflects temperature, depth encodes geometry, and X-ray reveals internal structure. Early VLM work has begun exploring non-RGB data. Chung et al. Chung et al. (2024) use GPT-4o to generate multiple-choice questions for

| Dataset | Temp | BBoxes | Seg | Reliability | Subjects | Primary Objective |
|---|---|---|---|---|---|---|
| Charlotte Ashrafi et al. (2022) | ✓ | ✗ | ✗ | ✗ | 10 | Facial thermography |
| M3FD Liu et al. (2022) | ✗ | ✓ | ✗ | ✗ | Several | Multi-modal object detection |
| LCAS Thermal Physio. Cosar et al. (2018) | ✓ | ✗ | ✗ | ✗ | 5 | Physiological monitoring |
| LCAS RGB-D-T Cosar & Bellotto (2019) | ✓ | ✓ | ✗ | ✗ | 15 | Human re-identification |
| FLIR ADAS FLIR (2024) | ✗ | ✓ | ✗ | ✗ | Several | ADAS object detection |
| LLVIP Zhu et al. (2021) | ✗ | ✓ | ✗ | ✗ | Several | Low-light pedestrian detection |
| SpeakingFaces Abdrakhmanova et al. (2021) | ✗ | ✓ | ✗ | ✗ | 142 | Speech and lipreading |
| Thermal Faces in the Wild Kuzdeuov et al. (2022) | ✗ | ✓ | ✗ | ✗ | 51 | Face and landmark recognition |
| **ThermEval (Ours)** | ✓ | ✓ | ✓ | ✓ | 25 | Vision Model Benchmarking |

Table 1: Comparison of thermal datasets, summarizing temperature data, bounding boxes, segmentation masks, annotator reliability, subject counts, and primary research objectives. ThermEval (Ours) provides comprehensive segmentation and annotator reliability across indoor and outdoor scenes.

multispectral images, including thermal, but rely on a single vision backbone and a constrained MCQ format, limiting generality and reasoning depth. RGB-Th-Bench Zhang et al. (2024b) studies RGB–thermal transfer but is restricted to binary tasks and does not evaluate temperature interpretation; its evaluation protocol also penalizes partially valid open-ended answers.

In contrast, **ThermEval** targets thermal-specific challenges through structured tasks spanning modality recognition, human counting, temperature reasoning, and per-pixel estimation. Unlike prior binary setups, our benchmark uses both classification and regression with quantitative metrics, enabling a more faithful and fine-grained assessment of VLM performance on thermal imagery

## 2.2 THERMAL AND INFRARED DATASETS

Several thermal datasets exist, but few expose temperature values. Widely used datasets such as FLIR_ADAS FLIR (2024), LLVIP Zhu et al. (2021), ThermalGAN Kniaz et al. (2018), and Mendely Ashfaq et al. (2021) have advanced multimodal perception research but lack the pixel-level temperature annotations needed for precise thermal reasoning. Only a few, including Charlotte-Faces Ashrafi et al. (2022) and the L-CAS Thermal Physiological Monitoring dataset Cosar et al. (2018), provide per-pixel temperature readings, though these are limited to facial imagery. The L-CAS RGBD-T dataset Cosar & Bellotto (2019) offers multimodal data but omits meaningful body-part annotations and focuses mainly on human re-identification. As summarized in Table 1, no dataset combines raw thermal imagery, per-pixel temperature maps, and diverse semantic contexts, a gap addressed by ThermEval-D.

## 2.3 FALSE-COLORED THERMAL IMAGES

Raw radiance or temperature matrices are rarely accessible; major datasets such as FLIR ADASFLIR (2024), LLVIPZhu et al. (2021), KAISTHwang et al. (2015), and OpenThermal-PoseKuzdeuov et al. (2025) release only false-colored thermal images. Consequently, false-colormapped imagery is the practical standard for downstream thermal analysis. Our approach follows established VLM practice, where sensor measurements are visualized before model ingestion. Prior work shows that VLMs learn reliably from such visualized physical modalities—including thermal and depthCai et al. (2025); Ashqar et al. (2024); Cao et al. (2025); Astrid et al. (2025); Huang et al. (2025). Although VLMs are predominantly trained on RGB, they are not restricted to it, and false-colored thermal images provide an effective interface for multimodal reasoning.

For **ThermEval**, we integrate FLIR ADAS and LLVIP for modality diversity, exclude datasets like ThermalGAN and Mendely that lack per-pixel ground truth, and introduce ThermEval-D to supply temperature-annotated thermal images with body-part labels missing in existing resources.

## 3 THERMEVAL

To evaluate vision-language models (VLMs) on thermal imagery, we introduce ThermEval-B, a suite of benchmark tasks testing perceptual and reasoning abilities, including modality identification, human counting, thermal reasoning, and temperature estimation. Each task uses a standardized prompt and is evaluated via an LLM-based judge or parser for consistent assessment. Details of the

data, evaluation methodology, and code are available in the repository here, with full implementation in Appendix B.2.

## 3.1 THERMEVAL-B: BENCHMARK

In this section, we provide an overview of the benchmark tasks.

**T1 Modality Identification:** The first task evaluates whether VLMs can recognize the visual characteristics of thermal imagery. We frame it as a modality classification problem using thermal–RGB image pairs from the FLIR and LLVIP datasets, with an equal distribution of RGB and thermal images. For each image, the VLM receives the prompt: *"Is this a thermal image or an RGB image?"*, and the ground truth corresponds to the actual modality of the image.

**T2 Modality Identification under Colormap Transformations:** This task extends T1 by testing whether VLMs can recognize thermal images when colorized with different colormaps. The prompt remains *"Is this a thermal image or an RGB image?"*. Colormaps enhance human interpretation of thermal data but alter appearance in ways that may confuse models. For example, Rainbow in medical diagnostics, Isotherm in industrial maintenance, and White Hot in law enforcement and wildlife tracking. Although the underlying thermal signal is unchanged, these transformations can shift model predictions. We evaluate performance on sequential colormaps (Type I, e.g., Magma and Viridis) and more complex colormaps (Type II, e.g., Summer and Spring), compared to standard grayscale representations Hunter (2024). The dataset is the same as T1, with colormap transformations applied to generate new images while retaining the thermal modality as ground truth.

**T3 Human Presence and Counting:** In Task 3, we evaluate a fundamental capability of VLMs: counting people in thermal images. Models receive the prompt: *"How many people are in this image? If there are no people, return 0."* We use thermal images from the FLIR and LLVIP datasets, which contain varying numbers of pedestrians in road scenes. Ground truth counts are determined from the annotated person labels in each image provided by FLIR and LLVIP datasets.

**T4 Reading the Colorbar:** This task evaluates whether VLMs can interpret the colorbar in thermal images, a prerequisite for temperature estimation and thermal reasoning. It consists of three components: (1) Colorbar detection, prompted with *"You are given a thermal image. Does it contain a color bar or temperature scale that maps colors to temperature values?"* to assess recognition of the colorbar's presence, with the colorbar absent in 50% of the images. (2) Colorbar localization, prompted with *"You are given a thermal image. It contains a color bar or temperature scale that maps colors to temperature values. What is the location of the colorbar?"* to identify its position (Top, Left, Bottom, Right). (3) Temperature range extraction, prompted with *"You are given a thermal image with a color bar or temperature scale that maps colors to temperature values. What is the maximum temperature value in degrees Celsius?"* to test interpretation of numerical values on the scale. Ground truth was programmatically generated by placing the colorbar in various locations.

**T5 Thermal Reasoning:** This task assesses VLMs' ability to reason about relative temperatures. It has two components: (1) Comparative reasoning across individuals, where images contain two people and models are prompted with *"Given the thermal image, determine whether the {body_part} of the left or right person is hotter. Respond with 'left' or 'right'."* Evaluated body parts include chest, forehead, and nose. (2) Within-individual reasoning, where images show a single person and models are asked *"Rank the following body parts from highest to lowest temperature: forehead, chest, nose."* The expected output is an ordered list reflecting actual thermal intensities. Ground truth was obtained from human annotations3.2.3, with the mean temperature of each body part used to determine correct ordering.

**T6 Temperature Estimation:** This task evaluates VLMs' ability to estimate temperatures from thermal images containing a colorbar. It has three levels of difficulty: (1) Coordinate-based estimation, where models are prompted *"Given the thermal image, what is the temperature at the coordinates ({x}, {y})? Return a single numerical value in degrees Celsius rounded to one decimal place (e.g., 17.6)."* (2) Pixel-based estimation, where models infer the temperature at a visually marked location, such as a red arrow. (3) Region-based estimation, with prompts like *"Given the thermal image, what is the temperature of the forehead of the right person? Return a single numerical value in degrees Celsius rounded to one decimal place (e.g., 17.6)."* Because thermal cameras measure skin surface temperature, which varies with ambient conditions, distance, and perspiration, accurate estimation

requires combining visual interpretation with reasoning over thermal properties. Ground truth for the first two subtasks was obtained programmatically using the known pixel locations, while for the region-based task it was derived from human annotations, using the mean temperature of each body part as the correct answer.

**T7 Temperature Estimation at Varying Depths:** This task evaluates how imaging distance affects VLMs' ability to estimate temperatures. We prompt models to predict the temperature of semantic regions such as the forehead or nose across three distances: 2ft, 6ft , and 10 ft. The prompt mirrors the region-based subtask in T6, for example: *"Given the thermal image, what is the temperature estimate of the forehead of the person according to the image? The temperature scale is in degrees Celsius. Please return a single numerical value rounded to one decimal place (e.g., 17.6)."* This setup enables systematic analysis of how depth impacts estimation accuracy and robustness.

### 3.2 THERMEVAL-D: DATASET

We present ThermEval-D, the first thermal image dataset covering both indoor and outdoor human-centric scenes with dense per-pixel temperature annotations. FLIR captures urban roads, LLVIP provides elevated street views, and ThermEval-D adds 500 images from everyday environments such as offices, parks, and workspaces. Each image includes detailed body-region annotations (forehead, chest, nose), enabling fine-grained tasks. By spanning diverse real-world contexts, ThermEval-D fills gaps in prior datasets and supports benchmarking of vision-language models across varied scenarios. Task-wise VQA counts are provided in Table 5. The dataset is available here[1].

#### 3.2.1 ETHICS STATEMENT

The study was approved by the Institutional Ethics Committee (IEC) under the protocol "Thermal Image Benchmarking for VLMs" (May 2025, six-month validity). Participants gave written consent, all personal data were anonymized, and the study adhered to institutional and national ethical standards. Any protocol changes or adverse events are reported to the IEC, and no study team members participated in the review.

#### 3.2.2 DATA COLLECTION PROTOCOL

We collected ThermEval-D across diverse indoor and outdoor environments within our institute, including offices, laboratories, workspaces, parks, and open grounds, following approval from the Institutional Ethics Committee (IEC). Twenty-five adult participants (age 18–47, weight 64–108 kg) with varied skin tones provided written consent and voluntarily participated. All procedures posed minimal risk, with the institute's medical center located 100 m from the sites. Participants performed natural activities such as standing, sitting, walking, and navigating stairs, allowing us to capture varied postures and thermal profiles.

We recorded thermal imagery using the TOPDON TC001 Plus camera, which features a $256 \times 192$ infrared sensor, <40 mK thermal sensitivity, 25 Hz frame rate, and a temperature range of –20°C to 550°C with ±1°C accuracy. We selected this commercially available camera because it provides per-pixel temperature annotations and reflects practical settings, as many applications cannot rely on high-end thermal equipment.

#### 3.2.3 DATASET ANNOTATION DETAILS

Each thermal image in ThermEval-D includes dense per-pixel temperature annotations, enabling fine-grained reasoning over spatial temperature patterns. Three expert annotators created polygonal segmentations following standardized guidelines with illustrative examples. Each image was annotated by all three annotators, and uncertainties were discussed collectively to ensure consistency across tasks. Bounding boxes were defined as follows:

**Person:** Encompasses the entire visible human body, including limbs, while excluding accessories.

**Forehead:** Extends from the hairline to the eyebrows, tightly cropped to avoid inclusion of eyes.

**Nose:** From bridge to nostrils, excluding adjacent facial regions; glasses were excluded unless thermally indistinguishable.

**Chest:** From base of neck to waistline, including shoulders and upper torso, excluding arms.

---

[1]https://tinyurl.com/ThermEval-Dataset

Bounding boxes were automatically derived from polygons for compatibility across tasks, supporting both coarse and fine spatial resolutions. Inter-annotator agreement, measured via IoU and Dice metrics, was strong (BBox IoU 0.77, Segm. IoU 0.72, BBox Dice 0.87, Segm. Dice 0.84), with pairwise agreements summarized in Table 4. For region-based tasks, ground truth temperatures were computed by averaging per-pixel values within segmentations (see Appendix A.1 for full data collection details).

## 4 EVALUATION

In this section, we detail the vision-language models used in our experiments and outline the evaluation protocol followed throughout the study. Please find implementation detail in Appendix B.

### 4.1 MODEL SPECIFICATIONS

We evaluated 15 open-source, 3 closed source and 3 chart finetuned vision-language models (VLMs) spanning diverse architectures, sizes, and origins, selected based on popularity and benchmark performance. This includes Intern-VL 3 (8B, 14B, 38B) Chen et al. (2024) and LLaVA 1.5 (7B) Xu et al. (2024), LLaMA 3.2 (11B) Grattafiori et al. (2024), MiniCPM-V 2.6 (8B) Yao et al. (2024), Phi-3 (4.2B), and Phi-3.5 (7B) Abdin et al. (2024), Qwen-VL (7B, 32B), Qwen-VL 2.5 (7B) , Qwen A22 (235B) Bai et al. (2023), PaliGemma-2 (3B) Steiner et al. (2024), IDEFICS-3 (6.7B) Laurençon et al. (2024), ,BLIP-2 (9B) Li et al. (2023) , Gemini 2.5 flash Team (2024), GPT-4o OpenAI (2024), Claude Haikuu Anthropic (2024), ChartGemma (3B) Masry et al. (2024b), ChartInstruct Llama-2(7B) Masry et al. (2024a) and TinyCharts (3B) Zhang et al. (2024a). This diverse set allows benchmarking across a wide spectrum of sizes and capabilities.

Focusing on open-source VLMs ensures reproducibility, transparency, and full access to weights and architectures, enabling rigorous evaluation and community-driven follow-up. Establishing strong zero-shot baselines on accessible models provides a foundation for comparison, fine-tuning, or adaptation, while yielding insights relevant to proprietary systems.

### 4.2 EVALUATION PROTOCOL

We evaluate all models in a strict zero-shot setting using a fixed prompt template, without any fine-tuning on thermal data. To ensure deterministic outputs, we set the decoding temperature to 0 and disable sampling. We restrict the maximum output length to 512 tokens, giving models sufficient capacity to reason and generate precise answers.

**LLM as a Judge:** Although we provide explicit formatting prompts, VLM outputs often vary structurally (Figure 1). Following prior work Danish et al. (2024); Zheng et al. (2023); Gu et al. (2024), we employ a language-only LLM judge (Gemini 2.5 models) to standardize and evaluate VLM predictions. In our pipeline, the judge receives as input the textual output of a VLM along with a few-shot prompt containing 3–5 examples, but it does not access the image itself. For classification tasks (T1, T2, T5), the judge outputs "Yes" if the VLM prediction matches the ground truth, and "No" otherwise. For regression tasks (T3, T4, T6, T7), it extracts numerical values from the VLM output to enable metric computation, such as mean absolute error. This approach ensures consistent evaluation across structurally diverse VLM outputs while leveraging the reasoning capabilities of a language model to parse text predictions. For regression tasks, our setup employs the LLM as a structured parser rather than as a scorer. Regex-based parsing was unreliable, while LLMs provided robust extraction, a trade-off also noted in prior work Gu et al. (2024). Stable decoding (temperature 0, sampling disabled) and task-specific few-shot prompts further ensured consistency.

**Benchmarking the Judge :** We validated our evaluation pipeline on a stratified gold set of 1,350 outputs spanning all tasks and models. Human annotators verified the correctness of the gold set, and we used structured judging with Gemini 2.5 Pro, Gemini 2.5 Flash, and Gemini 2.5 Flash Lite through the Instructor framework. These judges achieved agreement levels of 99.01 percent, 99.07 percent, and 98.24 percent respectively. Most remaining errors were due to ambiguous or truncated VLM responses rather than judge failures. The gold-set size was selected using standard statistical methods to ensure representativeness at the 95 percent confidence level with a margin of error below 3 percent. Appendix B.4 provides additional details.

Table 2: Comparison of VLM performance on **Task-1** (modality classification), **Task-2** (robustness to colormap transformations), **Task-3** (Human counting), and **Task-4** (Colorbar localisation and temperature extraction). ↑ indicates higher accuracy is better. ↓ indicates lower MAE is better. Text shown in red highlights comparatively lower performance among the models. **[NEW RESULTS ADDED]**

| Model | Params (in B) | Task-1 | | Task-2 | | Task-3 | | Task-4 | | | |
|---|---|---|---|---|---|---|---|---|---|---|---|
| | | FLIR ↑ | LLVIP ↑ | FLIR ↑ | LLVIP ↑ | FLIR ↓ | LLVIP ↓ | Detect ↑ | Position ↑ | Max ↓ | Min ↓ |
| SAM-3 | 0.9 | - | - | - | - | **2.07** | 0.66 | - | - | - | - |
| ChartGemma | 3.0 | 0.50 | 0.50 | 0.00 | 0.00 | 3.04 | 1.25 | 0.48 | 0.45 | 0.04 | 0.03 |
| TinyCharts | 3B | 0.50 | 0.50 | 0.00 | 0.00 | 4.72 | 2.99 | 0.5 | 0.14 | 68.44 | 24.75 |
| ChartInstruct Llama-2 | 7.0 | 0.50 | 0.50 | 0.00 | 0.01 | 4.48 | 2.36 | 0.5 | 0.25 | 162.08 | 74.37 |
| Phi-3 | 4.2 | 0.89 | 0.98 | 0.64 | 0.70 | 3.20 | 1.29 | **1.00** | 0.74 | **0.00** | **0.00** |
| IDEFICS-3 | 6.7 | 0.92 | 0.72 | 0.84 | 0.83 | 3.99 | 0.91 | **1.00** | 0.78 | **0.00** | 0.20 |
| LLaVA-1.5 | 7.0 | 0.97 | 0.89 | 0.89 | 0.72 | 3.43 | 1.22 | 0.50 | 0.31 | 11.00 | 2.51 |
| Phi-3.5 | 7.0 | 0.65 | 0.76 | 0.82 | 0.90 | 3.30 | 1.08 | **1.00** | 0.75 | **0.00** | **0.00** |
| Qwen-VL 2 | 7.0 | 0.97 | 0.99 | 0.99 | **1.00** | 3.65 | 0.75 | **1.00** | 0.73 | **0.00** | 2.05 |
| Qwen-VL 2.5 | 7.0 | 0.71 | 0.71 | 0.61 | 0.80 | 3.78 | 1.09 | **1.00** | 0.99 | **0.00** | 2.66 |
| Intern-VL 3 | 8.0 | **0.99** | **1.00** | **1.00** | **1.00** | 3.66 | 2.30 | **1.00** | **1.00** | 314.40 | 15.57 |
| MiniCPM-V 2.6 | 8.0 | 0.94 | 0.97 | 0.91 | 0.93 | 3.88 | 1.09 | **1.00** | 0.99 | **0.00** | **0.00** |
| BLIP-2 | 8.0 | 0.46 | 0.22 | 0.76 | 0.76 | 4.69 | 2.99 | 0.50 | 0.25 | 209.39 | 42.58 |
| PaliGemma-2 | 10.0 | 0.50 | 0.50 | 0.00 | 0.00 | 4.65 | 2.68 | 0.50 | 0.41 | 6.95 | 13.14 |
| LLaMA-3.2 | 11.0 | 0.98 | 0.86 | 0.77 | 0.63 | 2.88 | 0.70 | **1.00** | **1.00** | **0.00** | **0.00** |
| Intern-VL 3 | 14.0 | 0.96 | 0.99 | 0.86 | 0.97 | 2.79 | 0.73 | **1.00** | **1.00** | **0.00** | **0.00** |
| Qwen-VL 2.5 | 32.0 | 0.97 | 0.99 | 0.77 | 0.93 | 3.51 | 1.04 | **1.00** | **1.00** | 0.03 | 12.22 |
| Intern-VL 3 | 38.0 | **0.99** | **1.00** | **1.00** | **1.00** | 2.72 | **0.51** | **1.00** | **1.00** | **0.00** | **0.00** |
| Qwen-VL 2.5 (FT) | 7.0 | 1.00 | 1.00 | 1.00 | 1.00 | 1.85 | 0.55 | 1.00 | 1.00 | 0.00 | 0.01 |
| **Human** | – | 0.97 | 0.98 | 0.98 | 0.99 | **1.73** | **0.30** | **1.00** | **1.00** | **0.00** | **0.00** |
| Random Chance | – | 0.50 | 0.50 | 0.50 | 0.50 | – | – | 0.50 | 0.25 | – | – |

# 5 RESULTS

## 5.1 TASK 1 AND TASK 2: MODALITY IDENTIFICATION

Tasks 1 and 2 evaluate modality identification, with Task 2 adding colormap transformations as a robustness challenge (results in Table 2,). Human performance remained near perfect, with errors attributable to occasional mistakes. In Task 1, most VLMs perform strongly: Intern-VL 3 (38B) and Qwen-VL achieve near-human accuracy, indicating that distinguishing RGB from raw thermal images is relatively straightforward. Task 2 reveals substantial degradation: Intern-VL 3 (38B) remains robust, but PaliGemma-2, BLIP-2, and several Qwen-VL variants drop to near-random performance. Performance also varies by colormap type: sequential maps (Type I, e.g., Magma, Viridis) are more manageable, whereas complex maps (Type II, e.g., Summer, Spring) cause larger failures, suggesting reliance on low-level color statistics rather than modality-invariant features. Notably, PaliGemma consistently predicts RGB input, yielding fixed accuracies of 0.5 for Task 1 and 0 for Task 2. *Overall, while VLMs handle basic identification well, their robustness to colormap transformations is inconsistent.* This makes Task 2 a stronger diagnostic of true thermal modality understanding. Extensive results with colormap-specific performance are provided in Table 6.

## 5.2 TASK 3: HUMAN COUNTING

Task 3 evaluates VLMs' ability to detect human presence and accurately count individuals in thermal images. Results (Table 2) reveal wide variability across models. Early-generation systems such as BLIP-2, PaliGemma-2, and LLaVA-1.5 perform poorly, with MAE exceeding 3.4 on FLIR and 2.0 on LLVIP. In contrast, more recent models, including Qwen-VL, LLaMA-3.2, and Phi-3.5, achieve substantial improvements, reducing error to around 3 on FLIR and near 1 on LLVIP. Scaling trends are evident within the Intern-VL family: the 8B model struggles (MAE > 3.5 on FLIR), while the 14B and 38B variants improve markedly, with the 38B model reaching 2.72 on FLIR and 0.51 on LLVIP. Notably, Intern-VL (8B) exhibits a systematic failure, often defaulting to 11 when unable to resolve counts. Human annotators remain the most accurate, with MAE of 1.73 on FLIR and 0.3 on LLVIP. *Errors are most pronounced when images contain many individuals or overlapping thermal signatures, while both models and humans perform near-perfectly when counts are low or people are*

Table 3: Comparison of VLM performance on **Task-5** (Thermal reasoning), **Task-6** (Temperature estimation), and **Task-7** (Temperature estimation over varying depth). ↑ indicates higher accuracy is better. ↓ indicates lower MAE is better. Text in red highlights comparatively lower performance among the models. **x** indicates that model chose not to answer the question. **?** indicates that model parameters are unknown. **[NEW RESULTS ADDED]**

| Model | Params (in B) | Task-5 | | Task-6 | | | Task-7 | | |
|---|---|---|---|---|---|---|---|---|---|
| | | Double ↑ | Single ↑ | Coords ↓ | Arrow ↓ | Region ↓ | 2ft ↓ | 6ft ↓ | 10ft ↓ |
| ChartGemma | 3.0 | 0.52 | 0.27 | 5.43 | 5.91 | 5.43 | 4.44 | 3.56 | 3.25 |
| TinyCharts | 3.0 | 0.39 | | 13.81 | 5.19 | 5.25 | 3.31 | 3.09 | 2.85 |
| ChartInstruct Llama 2 | 7.0 | 0.00 | 0.28 | 32.61 | 14.01 | 6.01 | 3.09 | 3.18 | 3 .33 |
| Phi-3 | 4.2 | 0.57 | 0.27 | 6.02 | 6.34 | 4.58 | 5.82 | 6.18 | 6.74 |
| IDEFICS-3 | 6.7 | 0.47 | 0.38 | 5.91 | 5.89 | 4.41 | 2.35 | 2.22 | 2.58 |
| LLaVA-1.5 | 7.0 | 0.48 | 0.24 | 19.88 | 5.62 | 4.12 | 2.97 | 3.58 | 4.47 |
| Phi-3.5 | 7.0 | 0.42 | 0.28 | 5.65 | 5.83 | 3.59 | 2.15 | 2.29 | 2.56 |
| Qwen-VL 2 | 7.0 | 0.38 | 0.26 | 4.98 | 4.85 | 2.55 | 1.63 | 1.13 | 1.04 |
| Qwen-VL 2.5 | 7.0 | 0.41 | 0.42 | 3.65 | 4.75 | 2.91 | 1.05 | 1.00 | 1.00 |
| Intern-VL 3 | 8.0 | 0.41 | 0.34 | 80.95 | 31.48 | 11.15 | 6.49 | 16.59 | 20.30 |
| MiniCPM-V 2.6 | 8.0 | 0.40 | 0.27 | 4.00 | 6.32 | 4.28 | 2.15 | 2.03 | 1.85 |
| BLIP-2 | 8.0 | 0.39 | 0.16 | 13.08 | 12.74 | 14.73 | 16.96 | 16.35 | 15.43 |
| PaliGemma-2 | 10.0 | 0.44 | 0.00 | 6.39 | 5.67 | 7.80 | 6.29 | 5.38 | 4.59 |
| LLaMA-3.2 | 11.0 | 0.61 | 0.26 | 3.98 | 5.60 | 3.48 | 2.60 | 1.47 | 1.30 |
| Intern-VL 3 | 14.0 | 0.51 | 0.32 | 3.48 | 5.29 | 2.19 | 1.01 | 1.12 | 1.70 |
| Qwen-VL 2.5 | 32.0 | 0.43 | 0.33 | 7.67 | 8.74 | 2.95 | 1.54 | 1.66 | 1.97 |
| Intern-VL 3 | 38.0 | 0.50 | 0.37 | 9.92 | 4.61 | 1.76 | 1.57 | 1.54 | 1.73 |
| Qwen A22 | 235 B | 0.58 | 0.27 | 3.96 | 4.21 | 3.01 | 1.97 | 2.09 | 2.24 |
| Qwen-VL 2.5 (FT) | 7B | 0.58 | 0.56 | **1.58** | **1.55** | **1.03** | **0.53** | **0.49** | **0.61** |
| Gemini 2.5 flash | ?? | 0.54 | 0.28 | 3.81 | 3.48 | 2.50 | 1.30 | 1.80 | 1.96 |
| Claude Haiku 4.5 | ?? | 0.28 | **0.60** | 4.28 | 4.45 | 2.47 | 1.37 | 1.57 | 1.90 |
| GPT-4o | ?? | 0.46 | 0.34 | x | x | x | x | x | x |
| **Human** | – | **0.84** | 0.54 | – | **2.73** | 2.04 | 1.23 | 1.20 | 1.22 |
| Random Chance | – | 0.50 | 0.167 | – | – | – | – | – | – |

*well separated.*. This persistent gap, especially on FLIR, highlights the difficulty of robust human counting in thermal imagery and underscores it as a key open challenge for VLM-based reasoning.

### 5.3 TASK 4: COLORBAR INTERPRETATION

This task evaluates whether VLMs can interpret colorbars in thermal images, which is a prerequisite for downstream tasks such as temperature estimation. As shown in Table 2, nearly all modern VLMs, with the exception of PaliGemma 2, LLaVA 1.5, and BLIP 2, achieve near perfect accuracy in detecting the presence of a colorbar. Localization performance is also strong, with models such as Intern VL 3 (14B and 38B), LLaMA 3.2, and Qwen VL 2.5 (7B and 32B) reaching perfect accuracy. In contrast, PaliGemma 2, LLaVA 1.5, and BLIP 2 continue to struggle even with localization.

The main difficulty arises in extracting numerical temperature values. Only a few models, including Phi 3, Phi 3.5, MiniCPM V 2.6, LLaMA 3.2, and Intern VL 3 (14B and 38B), achieve zero error on both maximum and minimum temperature estimation. Others such as PaliGemma 2, BLIP 2, and LLaVA 1.5 produce errors greater than 2 to 6 °C. Scaling patterns are visible in the Intern VL 3 series: the 8B version produces very large errors (314.47 and 15.57 °C), while the 38B version eliminates them entirely. Some models exhibit systematic flaws. For example, BLIP 2 outputs only 0 or 273 for minimum and maximum values, and Intern VL 3 (8B) often shifts decimal points, reporting 334.2 °C instead of 33.42 °C. Humans remain perfectly accurate across all subtasks. These results reveal an important gap. While most VLMs can detect and localize colorbars, only a few can reliably interpret their numerical ranges. Moreover, *scaling model size alone does not ensure robustness in temperature extraction, which suggests that current architectures face a fundamental limitation*. Please refer to Table 8 for more details.

### 5.4 TASK 5: THERMAL REASONING

Task 5 evaluates VLMs' ability to reason over thermal intensities, beyond simple detection or localization. Performance lags sharply behind humans in both subtasks (Table 3). In the comparative reasoning setting with two people, accuracies range from 0.38 to 0.61, with LLaMA 3.2 performing

best, still well below the human benchmark of 0.84. Within-individual reasoning, which requires ranking body regions by thermal intensity, is even more challenging: most models score near random (0.24–0.38), with only Qwen VL 2.5 (7B) achieving 0.42 versus 0.54 for humans. Models such as PaliGemma 2 and BLIP 2 fail entirely. Scaling provides modest gains (e.g., Intern VL 3 improves from 0.41 at 8B to 0.51 at 38B) but cannot close the gap. These results highlight a fundamental limitation: *thermal reasoning demands structured relational understanding, not just larger model size*, underscoring the need for architectural innovation rather than parameter growth alone.

## 5.5 TASK 6: TEMPERATURE ESTIMATION

Task 6 evaluates VLMs' ability to estimate absolute temperatures from thermal images. Across coordinate- and arrow-based estimations, performance remains challenging: even the largest model, InternVL 3 (38B), achieves MAEs of 3.48°C (coordinate) and 4.61°C (arrow), still above the human baseline of 2.73°C. Smaller models, including InternVL 3 (8B), LLaVA, and BLIP-2, perform drastically worse (MAEs 80.95°C, 19.88°C, 13.08°C), reflecting their inability to map pixels to temperature values. The most striking failure is that some models, notably LLaVA, ignore the thermal image entirely, outputting fixed values (e.g., 37.5°C) for region-based estimation, effectively relying on language priors rather than visual inputs. Other models, such as PaliGemma 2 and BLIP-2, fail consistently across all subtasks. Region-based estimation proves more tractable: InternVL 3 (38B) achieves 1.76°C, surpassing humans, while Qwen-VL also performs competitively. These results reveal a systematic limitation: *many VLMs fail to ground predictions in thermal signals, defaulting to prior biases, and highlight the need for models designed to truly interpret and reason over thermal imagery*. Full results are in Table 9.

## 5.6 TASK 7: TEMPERATURE ESTIMATION AT VARYING DEPTH

Task 7 evaluates VLMs' ability to estimate temperatures from thermal images across different distances. Performance varies widely (Table 3). Early baselines such as BLIP-2 perform poorly (MAE >15 °C), indicating weak grounding in thermal inputs. Instruction-tuned models like Qwen-VL-2.5 and InternVL-14B achieve MAEs near 1 °C and remain stable across 2 ft, 6 ft, and 10 ft, demonstrating robust scaling behavior. In contrast, non-instruction-tuned InternVL exhibits a sharp degradation, with MAE increasing from 6.49 °C at 2 ft to 20.30 °C at 10 ft, revealing strong distance sensitivity and unreliable grounding. Mid-range models, including LLaVA and Phi, show moderate accuracy but gradual error increase with depth. These results highlight the critical importance of instruction tuning and model scale for reliable thermal reasoning across varying distances.

# 6 LIMITATIONS

Despite providing a comprehensive benchmark for VLMs on thermal imagery, our study has several limitations that also suggest future directions. We evaluate a limited subset of open-source and closed-source VLMs due to compute constraints, though ThermEval will expand to include more models. Second, we use a large language model as an automatic judge; while scalable, it can occasionally introduce minor errors, highlighting the value of enhanced automated checks. These limitations collectively point to opportunities for improving evaluation and advancing thermal reasoning research.

# 7 CONCLUSION

We present *ThermEval*, a comprehensive zero-shot benchmark and dataset with per-pixel temperature annotations for evaluating vision-language models on thermal imagery. Across a diverse set of classification and regression tasks, we reveal that current VLMs often fail to ground predictions in thermal signals, instead relying on language priors, showing systematic biases, or struggling with basic thermal reasoning. These results expose fundamental limitations of existing models and underscore the need for architectures and training strategies that truly integrate thermal modalities. By providing a rigorous evaluation framework, ThermEval lays the groundwork for developing thermal-aware multimodal models and advancing their deployment in real-world scenarios.

## 8 REPRODUCIBILITY STATEMENT

All the code and information regarding the experiments are available in the repository https://anonymous.4open.science/r/ThermEval. Additionally, Please find implementation detail in Appendix B.

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

# APPENDIX

## A DATASETS

### A.1 THERMEVAL-D DATASET

We release ThermEval-D, a high-resolution thermal image dataset with dense per-pixel temperature annotations, designed for tasks requiring precise temperature ground truths. The dataset contains over 500 images of human subjects, each annotated with detailed regions including the forehead, chest, nose, and full-body presence. All imagery was captured using the TOPDON TC001 Plus thermal camera, which features a 256×192 pixel infrared sensor, sub-40 mK thermal sensitivity, 25 Hz frame rate, and a temperature measurement range of –20°C to 550°C with ±1°C accuracy.

ThermEval-D addresses the scarcity of thermal datasets with dense temperature data in the research community. The complete dataset, along with its accompanying croissant metadata file, is publicly accessible via Kaggle here. A few sample of images from out Dataset are displayed in Figure 2 .

**Terms of Use and Licensing:** ThermEval-D is released under the Creative Commons Attribution-NonCommercial 4.0 (CC BY-NC 4.0) license, permitting unrestricted use for non-commercial research purposes.

**Data Maintenance and Accessibility:** The dataset is hosted on Kaggle, where we ensure long-term maintenance and periodic verification of accessibility. We plan regular expansions to enhance the dataset's scope and utility for the research community. Our benchmark involves ThermEval-D with other publicly available datasets for comprehensive evaluation across multiple tasks. While external datasets are used for comparative analysis, we do not redistribute them.

**ThermEval-D : Data Collection and Ethics:** Data collection was conducted across diverse settings within the authors' institution, including parks, open grounds, offices, laboratories, and workspaces, following approval from the Institutional Ethics Committee (IEC). The dataset includes participants from various demographic groups, covering different genders, age ranges, body types, and heights, all performing distinct activities with informed consent. This study was approved by the IEC under the protocol titled "Thermal Image Benchmarking for VLMs," valid from May 2025 for six months. All identifiable participant information was anonymized, and data collection posed minimal risk. Emergency medical support was readily available via the institutional medical center located approximately 100 meters from all collection sites.

**ThermEval-D Annotation Details :** Each image was annotated by three expert annotators who created polygonal segmentations following standardized guidelines. Bounding boxes were automatically derived from these polygons to maintain compatibility across tasks and allow both coarse and fine spatial resolution. Inter-annotator agreement was quantified using pairwise IoU and Dice metrics for both bounding boxes and polygons, with mean values of 0.77 (BBox IoU), 0.72 (Segm. IoU), 0.87 (BBox Dice), and 0.84 (Segm. Dice), reflecting strong consistency; for context, even a one-pixel shift in a $10{\times}10$ box yields IoU $\approx 0.68$, confirming that observed values indicate true agreement rather than noise. Temperature variability across annotators was assessed by calculating the standard deviation of per-pixel temperatures within each segmentation, yielding a representative image example of 32.26°C, 32.15°C, and 32.18°C (majority-vote 32.17°C, std 0.04°C), and a mean per-label standard deviation of 0.18°C across the dataset, demonstrating robust and reliable temperature extraction. These procedures ensure that ThermEval-D provides accurate, consistent, and reproducible annotations for both spatial and temperature-based evaluation tasks.

| Metric | Annotator Pairs | | | Mean of all Pairs |
|---|---|---|---|---|
| | 1 & 2 | 1 & 3 | 2 & 3 | |
| Bounding Box IoU | 0.7754 | 0.7477 | 0.7737 | 0.7656 |
| Segmentation IoU | 0.7248 | 0.7178 | 0.7308 | 0.7245 |
| Bounding Box Dice | 0.8735 | 0.8556 | 0.8724 | 0.8672 |
| Segmentation Dice | 0.8405 | 0.8357 | 0.8445 | 0.8402 |

Table 4: Inter-annotator agreement (IoU and Dice) for bounding boxes and segmentations across three annotators. The "Mean of Pairs" is the average of the three pairwise annotator scores.

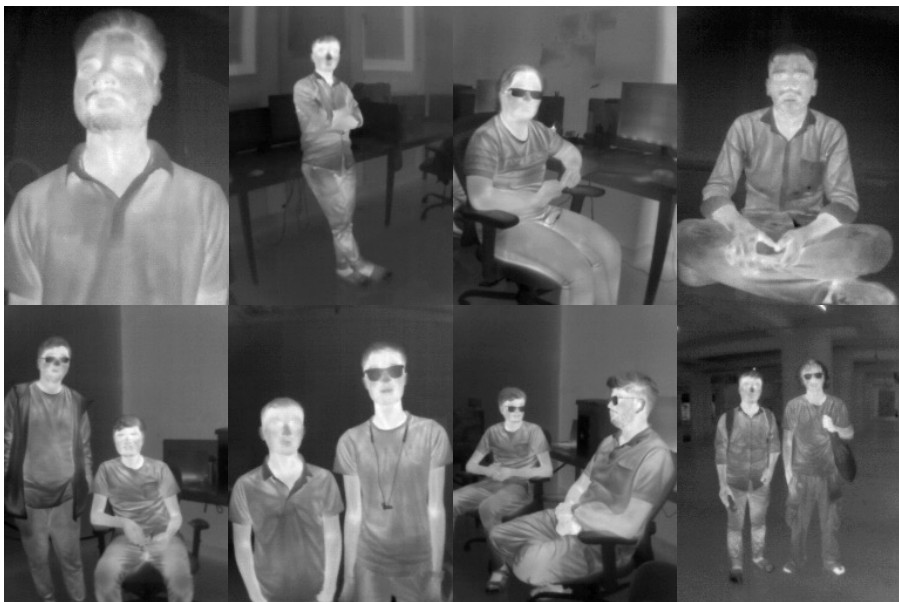

Figure 2: Images from ThermEval-D dataset. The top row shows the images having a single person in the scene whereas the second row shows the images having more than one person in the scene. Colorbars were added programatically during task evaluation

## A.2 FLIR-ADAS Dataset

The FLIR-ADAS dataset[2] is a publicly available resource (separate from the ThermEval-D dataset release) designed to advance research in thermal-visible fusion (RGBT) algorithms for autonomous driving applications. This dataset contains approximately 13,000 aligned thermal and RGB image pairs with multi-class annotations, including pedestrian labels; however, it lacks temperature annotations. The thermal images maintain a consistent resolution of 640×512 pixels, while RGB image resolutions vary throughout the dataset. Samples from the FLIR dataset are illustrated in Figure 3.

## A.3 LLVIP Dataset

The LLVIP dataset[3] is also a publicly available dataset (not a part of the ThermEval-D dataset release) that has thermal and RGB aligned images aimed at advancing fusion techniques for pedestrian detection in low-light conditions. It consists of about 15000 thermal RGB image pairs annotated with people. Both thermal and RGB images maintain uniform 1280×1024 pixel resolution. Notably, this dataset lacks per-pixel temperature annotations for thermal imagery. Sample images from the LLVIP dataset are presented in Figure 4.

---

[2]https://adas-dataset-v2.flirconservator.com/#downloadguide
[3]https://bupt-ai-cz.github.io/LLVIP/

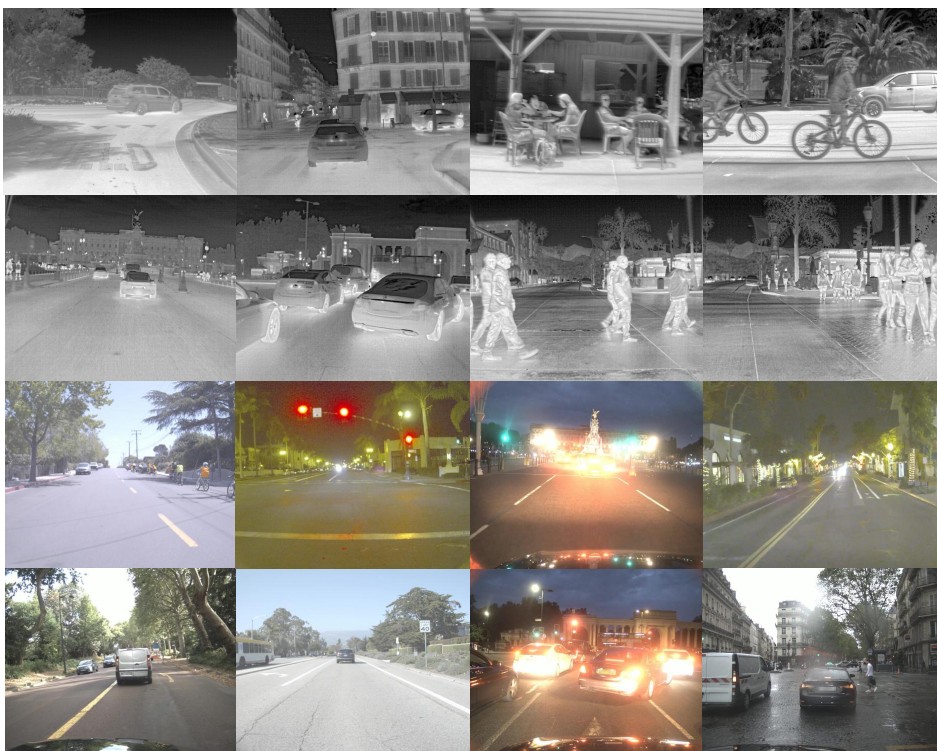

Figure 3: Demonstrates some of the images from the FLIR-ADAS dataset, which is used for Tasks T-1, T-2, and T-3. Top row shows thermal images while the bottom shows RGB for different scenes. More information regarding the tasks could be obtained from section 3.1.

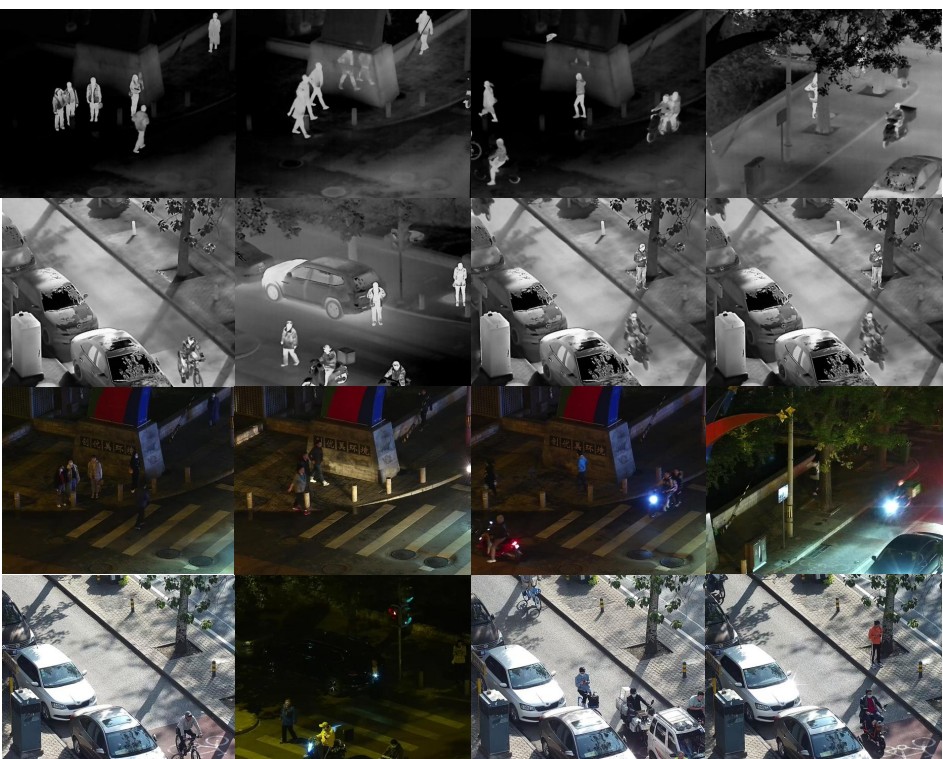

Figure 4: Demonstrates some of the images from the LLVIP dataset, which is used for Tasks T-1, T-2, and T-3.Top row shows thermal images while the bottom shows RGB for different scenes. More information regarding the tasks could be obtained from section 3.1.

Table 5: Number of VQA samples per task and dataset source in ThermEval benchmark.

| Task | Source | # VQA Samples |
|------|--------|---------------|
| T1 - Modality Identification | FLIR, LLVIP | 10,000 |
| T2 - Modality Identification (Colormap) | FLIR, LLVIP | 10,000 |
| T3 - Human Counting | FLIR, LLVIP | 20,000 |
| T4 - Colorbar (Double) | ThermEval-D | 156 |
| T4 - Colorbar (Single) | ThermEval-D | 145 |
| T5 - Thermal Reasoning (Arrow) | ThermEval-D | 2,400 |
| T5 - Thermal Reasoning (Coords) | ThermEval-D | 2,400 |
| T5 - Thermal Reasoning (Regions) | ThermEval-D | 717 |
| T6 - Temperature Estimation (Detection) | ThermEval-D | 480 |
| T6 - Temperature Estimation (Extraction) | ThermEval-D | 480 |
| T6 - Temperature Estimation (Max-Min) | ThermEval-D | 960 |
| T7 - Temperature Estimation at Depth (2 ft) | ThermEval-D | 248 |
| T7 - Temperature Estimation at Depth (6 ft) | ThermEval-D | 180 |
| T7 - Temperature Estimation at Depth (10 ft) | ThermEval-D | 138 |
| **Total** | – | 50,404 |

The FLIR-ADAS and LLVIP datasets were employed for Tasks T-1, T-2, and T-3, which evaluate fundamental VLM capabilities on thermal imagery without requiring specific temperature information. The ThermEval-D dataset was utilized for Tasks 4, 5 , 6 and 7, which necessitate precise temperature ground truth data for evaluation, a feature absent from existing publicly available thermal datasets.

## B  IMPLEMENTATION DETAILS

### B.1  COMPUTE SPECIFICATIONS

To ensure a fair comparison, all evaluations were conducted using the same hardware configuration: a single NVIDIA A100 GPU with 80GB of VRAM. Each evaluation involves a single forward pass (no ensembling or repeated sampling), and no access to model internals is assumed beyond what is publicly available through Hugging Face APIs or official released checkpoints. All prompt templates, prediction outputs, and evaluation scripts used in this study are provided in the accompanying GitHub repository.

### B.2  REPOSITORY STRUCTURE

The repository is accessible here. The root directory contains the following organizational structure:

1] **Datasets:** Contains all datasets utilized for model evaluation across different tasks.

2] **Evaluation:** Contains evaluation scripts for all tasks. These assess model performance across various tasks and saves the evalaution results, including the prompts used, correct answers, model outputs, and judge or parser outputs (task-dependent). Results are saved as a CSV separately for all the datasets.

3] **Evaluation Results:** Stores task-specific evaluation results as CSV files for all models across different datasets. These results are used for analysis and for arriving at results. All the results presented in the paper have been provided for transparency.

4] **Labels:** Contains task-specific ground truth labels saved as CSV files for model evaluation. These files include image paths and corresponding ground truth such as modality, colourmap used, person count, temperature at given coordinates, etc, and other task-relevant annotations.

5] **Processing Scripts:** Includes Python scripts designed for label generation. These scripts process the temperature matrices from the datasets folder alongside provided annotations to extract informa-

tion required for model evaluation. The processed information is stored as CSV files for different tasks within the Labels folder.

6] **Result Scripts:** Contains scripts for processing evaluation results and computing performance metrics. It also stores all plots and figures generated.

7] **Run.py:** The primary evaluation script for assessing vision-language models on all tasks. This script accepts model name as input parameter and saves evaluation results for the specified model in the evaluation results folder. To evaluate additional models not specified in this paper, users need to define the corresponding load_{model_name} and infer_{model_name} functions in the inference_model.py file located within the evaluation folder. Detailed instructions for this process are provided in the repository README.

### B.3 MODEL EVALUATION STEPS

**Setup:**

1] Download datasets: FLIR-ADAS (link), LLVIP (link), and ThermEval-D (link) from provided links.

2] Place the datasets (FLIR-ADAS, LLVIP and ThermEvalD) in the Datasets folder maintaining directory structure.

3] Create Python 3.8.10 virtual environment and install dependencies from requirements.txt.

**Execution:**

4] Run Run.py from root directory, specify model name [for example: 'llama', 'llava', 'phi', 'qwen_vl', 'minicpm', 'internvl']. Results are automatically saved to the evaluation results folder. Complete instructions are available in the repository README.

### B.4 SAMPLE SIZE JUSTIFICATION FOR LLM PARSER EVALUATION

To validate the LLM-based parser across all models and tasks, we created a gold set of approximately 1,200 parser outputs sampled from the full population of 700,000 outputs (50,000 VQA examples $\times$ 14 models). This sample size was chosen to provide statistically reliable estimates of parser accuracy while keeping annotation costs manageable.

Using the standard formula for finite-population proportions:

$$ n = \frac{Z^2 \, p \, (1 - p)}{e^2} \cdot \frac{N}{N - 1 + \frac{Z^2 \, p \, (1-p)}{e^2}} $$

where $n$ is the required sample size, $N = 700{,}000$ is the population of outputs, $p = 0.5$ is the conservative estimate for expected parser accuracy, $e = 0.03$ is the desired margin of error, and $Z = 1.96$ corresponds to a 95% confidence level, we obtain $n \approx 1{,}067$. This confirms that sampling approximately 1,200 outputs provides a 95% confidence interval of $\pm 3\%$ for proportion-based metrics such as exact match accuracy.

To ensure the gold set is representative, we performed stratified random sampling across tasks, models, and answer types, including edge cases such as multi-number outputs and malformed answers. This approach guarantees coverage of the full distribution of parser outputs, allowing us to estimate parser performance accurately for the entire population of 700,000 VLM outputs.

### B.5 ADDITIONAL RESULTS AND TASK-WISE IMPLEMENTATION DETAILS

#### B.5.1 TASK 1: MODALITY IDENTIFICATION

**Task:** This task aims to understand whether VLMs can visually distinguish RGB and Thermal Images.

**Prompt:** Is this a thermal image or an RGB image?

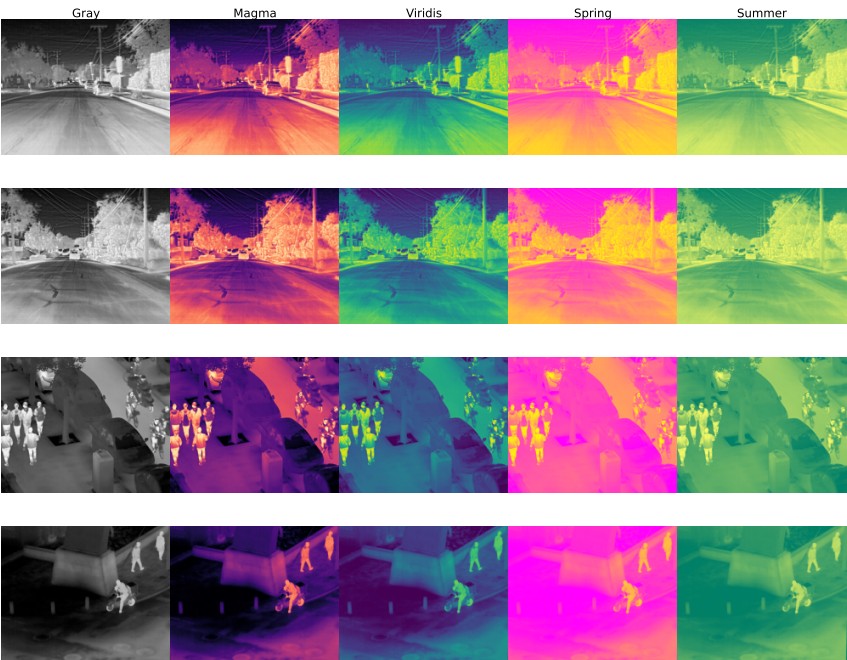

Figure 5: Demonstrates various colormaps used for Task T-2. Colormaps used were 'gray', 'magma', 'viridis', 'spring' and 'summer'

**Implementation Details:** This is a binary classification task, making its evaluation simple. We used 5,000 thermal-RGB image pairs each from the FLIR and LLVIP datasets, ensuring an equal number of thermal and RGB images for fair assessment. Sample images used for this task are shown in Figures 3 and 4.

### B.5.2 TASK 2: MODALITY IDENTIFICATION UNDER COLORMAP TRANSFORMATIONS

**Task:** This task extends task 1 by evaluating VLMS on thermal images with colormap transformations.

**Prompt:** Is this a thermal image or an RGB image?

**Implementation Details:** This is a binary classification task, making its evaluation simple. We used 1,000 thermal images each from the FLIR and LLVIP datasets, applying five colormap transformations per image to create a total of 10,000 images. We used simple sequential colormaps (Type I) such as Magma and Viridis, and more complex ones (Type II) like Summer and Spring, along with standard grayscale thermal images. Sample images used in this task are shown in Figures 5.

### B.5.3 TASK 3: COUNTING HUMANS

**Task:** This task assesses the basic object counting capability of VLMS, specifically focusing on counting people.

**Prompt:** How many people are in this image? If there are no people, return 0.

**Implementation Details:** This regression task used 10,000 grayscale thermal images each from the FLIR and LLVIP datasets. A separate model parsed the outputs to estimate the numerical count of people.

### B.5.4 TASK 4: READING COLORBAR

**Task:** This task evaluates the VLMs ability to identify and read the colorbar. It comprises of 3 subtasks (a) Identifying the presence of colorbar, (b) Identifying the location of the colorbar (top, left, bottom or right), and (c) Extracting the max and min value on the Colorbar.

| Model | Params (B) | Gray | | Magma | | Spring | | Summer | | Viridis | |
|---|---|---|---|---|---|---|---|---|---|---|---|
| | | FLIR | LLVIP | FLIR | LLVIP | FLIR | LLVIP | FLIR | LLVIP | FLIR | LLVIP |
| Phi-3 | 4.2 | 0.80 | 0.99 | 0.99 | 0.93 | 0.40 | 0.55 | 0.06 | 0.06 | 0.97 | 0.98 |
| IDEFICS 3 | 6.7 | 0.92 | 0.54 | 1.00 | 0.99 | 0.42 | 0.78 | 0.89 | 0.98 | 0.99 | 0.87 |
| LLaVA-1.5 | 7.0 | 0.94 | 0.83 | 1.00 | 0.93 | 1.00 | 0.77 | 0.53 | 0.35 | 1.00 | 0.70 |
| Phi-3.5 | 7.0 | 0.31 | 0.55 | 1.00 | 0.98 | 0.99 | 1.00 | 0.81 | 0.99 | 1.00 | 0.99 |
| Qwen-VL | 7.0 | 0.96 | 0.99 | 1.00 | 1.00 | 1.00 | 1.00 | 1.00 | 1.00 | 1.00 | 1.00 |
| MiniCPM-V 2.6 | 8.0 | 0.88 | 0.95 | 1.00 | 1.00 | 0.87 | 0.91 | 0.82 | 0.80 | 1.00 | 1.00 |
| InternVL | 8.0 | 1.00 | 1.00 | 1.00 | 1.00 | 1.00 | 1.00 | 1.00 | 1.00 | 1.00 | 1.00 |
| BLIP-2 | 9.0 | 0.58 | 0.53 | 0.83 | 0.75 | 0.87 | 0.99 | 0.99 | 0.93 | 0.58 | 0.64 |
| PaliGemma-2 | 10.0 | 0.00 | 0.00 | 0.00 | 0.00 | 0.00 | 0.00 | 0.00 | 0.00 | 0.00 | 0.00 |
| LLaMA-3.2 | 11.0 | 0.97 | 0.63 | 1.00 | 1.00 | 0.90 | 0.60 | 0.01 | 0.00 | 0.99 | 0.94 |
| InternVL | 14.0 | 0.91 | 1.00 | 0.84 | 0.99 | 0.89 | 0.99 | 0.70 | 0.93 | 0.99 | 0.96 |
| Qwen-VL 2.5 | 32.0 | 0.74 | 0.96 | 1.00 | 1.00 | 0.77 | 0.89 | 0.40 | 0.82 | 0.97 | 1.00 |
| InternVL | 38.0 | 1.00 | 1.00 | 1.00 | 1.00 | 1.00 | 1.00 | 0.99 | 1.00 | 1.00 | 1.00 |

Table 6: Accuracy of VLMS on Task-2: Modality Identification under colormap transformation with results shown separately for FLIR and LLVIP datasets. Higher numbers are better.

| Model | Params (B) | FLIR | | | | LLVIP | | | |
|---|---|---|---|---|---|---|---|---|---|
| | | MAE ↓ | STD ↓ | Bias * | RMSE ↓ | MAE ↓ | STD ↓ | Bias * | RMSE ↓ |
| Phi-3 | 4.2 | 3.20 | 4.24 | -3.12 | 5.26 | 1.29 | 1.25 | -1.22 | 1.75 |
| Phi-3.5 | 7.0 | 3.30 | 4.49 | -3.23 | 5.53 | 1.08 | 1.15 | -1.01 | 1.53 |
| IDEFICS-3 | 6.7 | 3.99 | 5.31 | -3.98 | 6.63 | 0.91 | 1.13 | -0.74 | 1.35 |
| LLaVA-1.5 | 7.0 | 3.43 | 4.75 | -3.33 | 5.80 | 1.22 | 1.56 | -0.92 | 1.81 |
| Qwen-VL | 7.0 | 3.65 | 5.11 | -3.63 | 6.27 | 0.75 | 1.09 | -0.33 | 1.14 |
| MiniCPM-V 2.6 | 8.0 | 3.88 | 4.99 | -3.87 | 6.31 | 1.09 | 1.32 | -0.98 | 1.65 |
| Intern-VL 3 | 8.0 | 3.66 | 5.42 | -1.44 | 5.61 | 2.30 | 3.86 | 1.64 | 4.20 |
| BLIP-2 | 9.0 | 4.69 | 5.59 | -4.69 | 7.30 | 2.99 | 1.82 | -2.99 | 3.50 |
| PaliGemma-2 | 10.0 | 4.65 | 5.59 | -4.65 | 7.27 | 2.68 | 1.88 | -2.65 | 3.25 |
| LLaMA-3.2 | 11.0 | 2.88 | 4.05 | -2.72 | 4.88 | 0.70 | 1.04 | -0.21 | 1.07 |
| Intern-VL 3 | 14.0 | 2.79 | 4.15 | -2.76 | 4.98 | 0.73 | 1.01 | -0.59 | 1.17 |
| Qwen-VL 2.5 | 32.0 | 3.51 | 4.77 | -3.49 | 5.91 | 1.04 | 1.20 | -0.91 | 1.51 |
| Intern-VL 3 | 38.0 | 2.72 | 4.08 | -2.69 | 4.88 | 0.51 | 0.82 | -0.30 | 0.88 |

Table 7: Regression metrics for Task-3 : Human Counting using FLIR and LLVIP datasets. ↓ indicates lower is better. *Bias closer to 0 is better.

**Prompt 1:** You are given a thermal image. Does it contain a color bar or temperature scale that maps colors to temperature values? Answer only with 'Yes' or 'No'.

**Prompt 2:** You are given a thermal image. It contain a color bar or temperature scale that maps colors to temperature value. What is the location of the colorbar? Possible locations are top, left, bottom, right.

**Prompt 3:** You are given a thermal image with a color bar or temperature scale that maps colors to temperature value. What is the maximum temperature value in degree Celsius?

**Implementation Details:** This task contains both classification as well as regression task. Prompt 1 and 2 would lead to a classification task where as the task 3 would lead to regression task. From subtask (a) the random chance accuracy is 50% whereas for subtask (b) the random chance accuracy is 25%.

### B.5.5    TASK 5: TEMPERATURE REASONING

**Task:** This task evaluates the reasoning capabilities of VLMS in thermal domain. It comprises of 2 subtasks: (a) Ranking the chest, head and nose of a person from hottest to coldest and (b) To compare the temperature head/chest/nose of 2 people in the image and return "left" or "right".

| Model | Params (B) | Detection | Position | Extraction | | | | | |
|---|---|---|---|---|---|---|---|---|---|
| | | Accuracy | Accuracy | Acc Max | Acc Min | Acc | MAE Max | MAE Min | MAE |
| BLIP-2 | 9.0 | 0.50 | 0.25 | 0.00 | 0.00 | 0.00 | 209.39 | 42.58 | 68.42 |
| IDEFICS-3 | 6.7 | 1.00 | 0.78 | 1.00 | 1.00 | 1.00 | 0.00 | 0.20 | 0.10 |
| Intern-VL 3 | 8.0 | 1.00 | 1.00 | 0.30 | 0.88 | 0.59 | 314.40 | 15.57 | 163.40 |
| Intern-VL 3 | 14.0 | 1.00 | 1.00 | 1.00 | 1.00 | 1.00 | 0.00 | 0.00 | 0.00 |
| Intern-VL 3 | 38.0 | 1.00 | 1.00 | 1.00 | 1.00 | 1.00 | 0.00 | 0.00 | 0.00 |
| LLaMA-3.2 | 11.0 | 1.00 | 1.00 | 1.00 | 0.99 | 0.99 | 0.00 | 0.00 | 0.00 |
| LLaVA-1.5 | 7.0 | 0.50 | 0.31 | 0.01 | 0.18 | 0.10 | 11.00 | 2.51 | 6.76 |
| MiniCPM-V 2.6 | 8.0 | 1.00 | 0.99 | 1.00 | 1.00 | 1.00 | 0.00 | 0.00 | 0.00 |
| PaliGemma-2 | 10.0 | 0.50 | 0.41 | 0.19 | 0.21 | 0.20 | 6.95 | 13.14 | 10.04 |
| Phi-3 | 4.2 | 1.00 | 0.74 | 1.00 | 1.00 | 1.00 | 0.00 | 0.00 | 0.00 |
| Phi-3.5 | 7.0 | 1.00 | 0.75 | 1.00 | 1.00 | 1.00 | 0.00 | 0.00 | 0.00 |
| Qwen-VL | 7.0 | 1.00 | 0.73 | 0.99 | 0.95 | 0.97 | 0.00 | 2.05 | 1.02 |
| Qwen-VL 2.5 | 32.0 | 1.00 | 0.99 | 1.00 | 0.94 | 0.97 | 0.00 | 2.66 | 1.33 |

Table 8: Model evaluation for Task-4: colorbar interpretation task, assessing the ability to detect, position, and extract temperature values. Acc Max and Acc Min denotes the accuracy of correctly identifying maximum and minimum values of the colorbar. MAE Max and MAE Min denotes the MAE is estimating Max and Min temperature of the colorbar.

**Prompt 1:** Given the thermal image, determine whether the {body part} of the left or right person is hotter. Respond with 'left' or 'right'.

**Prompt 2:** Rank the following body parts from highest to lowest temperature: head, chest, nose.

**Implementation Details:** This task involves binary classification and ordering, using the ThermEval-D dataset as it requires the temperature ground truths. The thermal image of size 256 x 192 which is same as the size of the temperature matrix, ie, 256 x 192, and mean temperatures were computed for regions defined by polygon box coordinates.

### B.5.6 TASK 6: TEMPERATURE ESTIMATION

**Task:** This task analyzes the model's ability to estimate the temperature of given pixels or regions using the colorbar in the image. It is sub-divided into 3 subtasks- (a) Given the coordinates, the model is prompted to estimate the temperature of the given pixel, (b) The model is prompted to estimate the temperature of the pixel marked by a red arrow and (c) The model is required to estimate the temperature of semantic regions like the head, chest or the nose.

**Prompt 1:** Given the thermal image, what is the temperature at the coordinates ({x},{y})? The temperature scale is in degrees Celsius. Please return a single numerical value rounded to one decimal place (e.g., 17.6).

**Prompt 2:** Given the thermal image, what is the temperature at the point marked by the red arrow? The temperature scale is in degrees Celsius. Please return a single numerical value rounded to one decimal place (e.g., 17.6).

**Prompt 3.1:** Given the thermal image, what is the temperature estimate of the {body_part} according to the image? The temperature scale is in degrees Celsius. Please return a single numerical value rounded to one decimal place (e.g., 17.6).

**Prompt 3.2:** Given the thermal image, what is the temperature estimate of the {body_part} of the {right/left} person according to the image? The temperature scale is in degrees Celsius. Please return a single numerical value rounded to one decimal place (e.g., 17.6).

**Implementation Details:** All three subtasks are regression tasks using the ThermEval-D dataset, with temperature ground truths obtained via mean of polygon segmentation of temperatures. For the first two subtasks, the coordinates were generated randomly, constrained to the central region of the images to avoid excessive background representation or overlapping with the temperature scale. In the second subtask, the angle of the red arrow marking the pixel was also randomized.

| Model | Params (B) | Arrow | | | | Coordinates | | | | Region | | | |
|---|---|---|---|---|---|---|---|---|---|---|---|---|---|
| | | MAE↓ | RMSE↓ | BIAS* | STD↓ | MAE↓ | RMSE↓ | BIAS* | STD↓ | MAE↓ | RMSE↓ | BIAS* | STD↓ |
| IDEFICS 3 | 6.7 | 5.89 | 7.36 | 3.07 | 6.69 | 5.91 | 7.13 | 4.61 | 5.44 | 4.41 | 5.81 | 1.93 | 5.48 |
| LLaVA-1.5 | 7.0 | 5.62 | 6.94 | 4.54 | 5.25 | 19.88 | 69.80 | 13.88 | 69.77 | 4.12 | 4.90 | 3.96 | 2.87 |
| Phi-3.5 | 7.0 | 5.83 | 6.89 | 4.20 | 5.46 | 5.65 | 6.75 | 4.03 | 5.41 | 3.59 | 4.13 | 2.77 | 3.07 |
| Qwen-VL | 7.0 | 4.85 | 6.19 | 3.94 | 4.78 | 4.98 | 6.25 | -2.22 | 5.84 | 2.55 | 3.35 | 2.08 | 2.62 |
| MiniCPM-V 2.6 | 8.0 | 6.32 | 7.48 | 3.00 | 6.85 | 4.00 | 5.29 | -1.97 | 4.91 | 4.28 | 5.43 | 1.42 | 5.24 |
| InternVL | 8.0 | 31.48 | 92.95 | 29.46 | 88.16 | 80.95 | 152.63 | 80.12 | 129.91 | 11.15 | 130.18 | 9.97 | 129.79 |
| BLIP-2 | 9.0 | 12.74 | 13.17 | -12.74 | 3.35 | 13.08 | 15.08 | -12.34 | 8.66 | 14.73 | 14.89 | -14.73 | 2.15 |
| PaliGemma-2 | 10.0 | 5.67 | 6.93 | -5.65 | 4.02 | 6.39 | 11.75 | -4.80 | 10.72 | 7.80 | 8.58 | -7.78 | 3.63 |
| LLaMA-3.2 | 11.0 | 5.60 | 6.74 | 1.70 | 6.52 | 3.98 | 5.26 | 1.84 | 4.92 | 3.48 | 4.95 | -0.96 | 4.86 |
| InternVL | 14.0 | 5.29 | 6.41 | 1.12 | 6.31 | 3.48 | 4.43 | 1.69 | 4.10 | 2.19 | 2.85 | 1.09 | 2.63 |
| Qwen-VL 2.5 | 32.0 | 4.75 | 5.98 | 1.66 | 5.74 | 3.65 | 4.71 | 0.48 | 4.68 | 2.91 | 3.59 | 2.09 | 2.92 |
| Qwen-VL 2.5 | 32.0 | 8.74 | 16.09 | -2.56 | 15.89 | 7.67 | 15.90 | -3.75 | 15.45 | 2.95 | 4.79 | 2.22 | 4.25 |
| InternVL | 38.0 | 4.61 | 5.80 | 0.70 | 5.75 | 9.92 | 16.32 | 8.90 | 13.68 | 1.76 | 2.28 | 1.14 | 1.98 |

Table 9: Regression metrics for Task-6: Temperature Estimation on ThermEval Dataset. ↓ indicates lower is better. *Bias closer to 0 is better.

| Model | Params (B) | 2ft | | | | 6ft | | | | 10ft | | | |
|---|---|---|---|---|---|---|---|---|---|---|---|---|---|
| | | MAE↓ | RMSE↓ | BIAS* | STD↓ | MAE↓ | RMSE↓ | BIAS* | STD↓ | MAE↓ | RMSE↓ | BIAS* | STD↓ |
| phi | 4.2 | 5.82 | 14.21 | -2.72 | 13.94 | 6.18 | 14.94 | -2.33 | 14.75 | 6.74 | 15.30 | -2.78 | 15.04 |
| IDEFICS 3 | 6.7 | 2.35 | 2.67 | 1.42 | 2.26 | 2.22 | 2.48 | 1.55 | 1.93 | 2.58 | 2.85 | 1.28 | 2.55 |
| LLaVA-1.5 | 7.0 | 2.97 | 3.21 | 2.97 | 1.20 | 3.58 | 3.84 | 3.58 | 1.38 | 4.47 | 4.67 | 4.47 | 1.37 |
| Phi-3.5 | 7.0 | 2.15 | 2.40 | 1.82 | 1.56 | 2.29 | 2.48 | 2.16 | 1.22 | 2.56 | 2.73 | 2.56 | 0.95 |
| Qwen-VL | 7.0 | 1.63 | 1.85 | -0.30 | 1.83 | 1.13 | 1.36 | -0.12 | 1.36 | 1.04 | 1.29 | -0.26 | 1.26 |
| Qwen-VL 2.5 | 7.0 | 1.05 | 1.33 | 0.19 | 1.32 | 1.00 | 1.22 | 0.60 | 1.06 | 1.00 | 1.21 | 0.62 | 1.03 |
| MiniCPM-V 2.6 | 8.0 | 2.15 | 2.42 | 0.53 | 2.36 | 2.03 | 2.31 | 0.58 | 2.23 | 1.85 | 2.18 | 0.93 | 1.97 |
| InternVL | 8.0 | 6.49 | 38.49 | 5.67 | 38.07 | 16.59 | 67.33 | 16.34 | 65.31 | 20.30 | 75.45 | 20.24 | 72.68 |
| BLIP-2 | 9.0 | 16.96 | 17.00 | -16.96 | 1.11 | 16.35 | 16.40 | -16.35 | 1.28 | 15.43 | 15.50 | -15.43 | 1.37 |
| PaliGemma-2 | 10.0 | 6.29 | 6.45 | -6.29 | 1.42 | 5.38 | 5.45 | -5.38 | 0.85 | 4.59 | 4.66 | -4.59 | 0.84 |
| LLaMA-3.2 | 11.0 | 2.60 | 3.12 | -1.78 | 2.56 | 1.47 | 1.79 | -0.81 | 1.59 | 1.30 | 1.66 | -0.47 | 1.59 |
| InternVL | 14.0 | 1.01 | 1.27 | 0.66 | 1.09 | 1.12 | 1.38 | 0.94 | 1.01 | 1.70 | 1.96 | 1.59 | 1.14 |
| Qwen-VL 2.5 | 32.0 | 1.54 | 1.88 | 1.34 | 1.32 | 1.66 | 1.89 | 1.59 | 1.01 | 1.97 | 2.23 | 1.95 | 1.08 |
| InternVL | 38.0 | 1.57 | 1.80 | 1.15 | 1.38 | 1.54 | 1.80 | 1.34 | 1.21 | 1.73 | 2.00 | 1.61 | 1.19 |

Table 10: Regression metrics for Task-7: Temperature Estimation at varying depth on ThermEval dataset. ↓ indicates lower is better. *Bias closer to 0 is better.

### B.5.7 TASK 7: TEMPERATURE ESTIMATION AT VARYING DISTANCE

**Task:** This task analyzes the model's ability to estimate the temperature of given pixels or regions using the colorbar in the image. unlike previous task it is devided by varying distances of 1m , 4m and 6. The model is required to estimate the temperature of semantic regions like the head, chest or the nose.

**Prompt 3.2:** Given the thermal image, what is the temperature estimate of the {body_part} of the {right/left} person according to the image? The temperature scale is in degrees Celsius. Please return a single numerical value rounded to one decimal place (e.g., 17.6).

**Implementation Details:** Same as that of Task-6.

### B.6 SUPERVISED FINETUNING EXPERIMENT

### B.6.1 EXPERIMENTAL SETUP

We fine tuned Qwen2.5 VL 7B Instruct for 5 epochs using LoRA with rank $(r = 16)$ and scaling factor $(\alpha = 16)$, applied to the query, key, value, and output projection layers with dropout (0.1). We used Paged AdamW 32 bit with a fixed learning rate $(5 \times 10^{-6})$, no warmup, batch size 4 per device. The dataset was split into three stratified folds to balance tasks and subtasks, ensuring each VQA sample was seen exactly once and enabling full dataset evaluation without repetition.

### B.6.2 FINDINGS

Our results show that finetuning Qwen-VL 2.5 (7B) enables the model to outperform the much larger Qwen A22 235B and all other evaluated open- and closed-source VLMs. The finetuned model

Table 11: Comparison of finetuned Qwen-VL 2.5 (7B) with human performance and other models. ↓ indicates lower is better and ↑ indicates higher is better. "–" indicates not applicable.

| Task | Best Model (Zero-shot) | Human | Qwen 2.5 7B Zero-shot | Qwen 2.5 7B Finetuned | Qwen Δ |
|---|---|---|---|---|---|
| T1 FLIR (Acc ↑) | 1.00 | 0.97 | 0.71 | **1.00** | +0.29 |
| T1 LLVIP (Acc ↑) | 1.00 | 0.98 | 0.71 | **1.00** | +0.29 |
| T2 FLIR (Acc ↑) | 1.00 | 0.98 | 0.61 | **1.00** | +0.39 |
| T2 LLVIP (Acc ↑) | 1.00 | 0.98 | 0.80 | **1.00** | +0.20 |
| T3 FLIR (MAE ↓) | 2.72 | **1.73** | 3.78 | 1.85 | −1.93 |
| T3 LLVIP (MAE ↓) | 0.51 | **0.30** | 1.09 | 0.55 | −0.54 |
| T4 Detect (Acc ↑) | 1.00 | **1.00** | 1.00 | **1.00** | 0.00 |
| T4 Position (Acc ↑) | 1.00 | **1.00** | 0.99 | **1.00** | +0.01 |
| T4 Max (MAE ↓) | 0.00 | **0.00** | 0.00 | **0.00** | 0.00 |
| T4 Min (MAE ↓) | 0.00 | **0.00** | 2.66 | **0.00** | −2.66 |
| T5 Double (Acc ↑) | 0.61 | **0.84** | 0.41 | 0.58 | +0.17 |
| T5 Single (Acc ↑) | **0.60** | 0.54 | 0.42 | 0.56 | +0.14 |
| T6 Cords (MAE ↓) | 3.48 | – | 3.65 | **1.58** | −2.07 |
| T6 Arrow (MAE ↓) | 3.48 | 2.73 | 4.75 | **1.55** | −3.20 |
| T6 Region (MAE ↓) | 1.76 | 2.04 | 2.91 | **1.03** | −1.88 |
| T7 2ft (MAE ↓) | 1.01 | 1.23 | 1.05 | **0.53** | −0.52 |
| T7 6ft (MAE ↓) | 1.00 | 1.20 | 1.00 | **0.49** | −0.51 |
| T7 10ft (MAE ↓) | 1.00 | 1.22 | 1.00 | **0.61** | −0.39 |

matches or exceeds human performance on most tasks, demonstrating that targeted supervision is highly effective for thermal reasoning. These results indicate that ThermEval provides meaningful, domain-grounding supervision and establishes a reliable benchmark for advancing thermal understanding in VLMs.

- **Finetuning improves performance but does not solve thermal reasoning.** SFT reduces MAE and improves accuracy, indicating that VLMs have the latent capacity to handle thermal data. However, the remaining errors are still substantial: absolute temperature estimates deviate by 1–2 °C in T6/T7, and performance on semantic comparison tasks (T5) remains below human-level. For applications such as fever screening or industrial hotspot detection, such error margins are not acceptable and highlight the need for deeper physical grounding. For instance, a standard deviation of 1-2 °C would be unsuitable for any model intended for non-contact fever detection.
- **SFT closes the gap because current VLMs lack domain grounding, not capacity.** VLMs do not inherently understand temperature as a physical quantity or thermal appearance as a modality, even though they can acquire these concepts with minimal supervision. The fact that small-scale finetuning resolves failures on basic tasks suggests that the primary limitation lies in incomplete training signals rather than in model architecture or scale.
- **ThermEval isolates primitive abilities that RGB-centric pretraining does not teach.** Pretrained VLMs, which are predominantly exposed to RGB photographs, diagrams, and charts, tend to learn mappings from appearance to semantic categories. Thermal understanding, however, requires mapping appearance to a physical quantity such as temperature. Because current models are not trained on this type of signal, they do not naturally acquire it, and SFT can only partially bridge this gap.
- **Future VLM pretraining should include physical sensor modalities.** Most existing VLMs are trained primarily on RGB imagery, and although the training data of closed-source models are not public, available evidence and model behavior suggest limited exposure to thermal infrared data. This likely contributes to why current models interpret thermal images as RGB-like visuals rather than as physical measurements. Recent efforts, such as the Gemini team's inclusion of modalities like X-rays and CT scans, and similar advances in remote-sensing VLMs, demonstrate that expanding pretraining beyond RGB is both feasible and beneficial. These developments indicate that incorporating additional physical sensing modalities is an important direction for future VLM development.

### B.7 EXPERIMENTING WITH PROMPT ABLATIONS

In these experiments, we added contextual information about the thermal images to guide the VLM toward the relevant aspects of the scene. We conducted a systematic study across Intern-VL (14B), MiniCPM (8B), Qwen-VL-2.5 (7B), and BLIP2 (9B), comparing the original zero-shot prompts with context-augmented versions.

#### B.7.1 EXAMPLES OF ABLATION MADE

We describe below the changes we introduced to evaluate the effect of prompt ablations.

**Task T1,T2**

- **Original:** "Is this a thermal image or an RGB image?"
- **Ablation:** "RGB images come from visible light and depict natural color and texture. Each pixel represents the intensity of red, green and blue channels that together form the visual appearance of objects under illumination. Thermal images measure emitted infrared radiation and encode temperature dependent signals. Each pixel represents a temperature value or a value proportional to heat emission, and any colors seen in the image come from a colormap applied to these underlying temperature readings Based on above context is the given image an RGB or Thermal Image?"

**Task T3**

- **Original:** "Count the number of humans in the image."
- **Ablation:** "You are given a thermal images. Thermal images measure emitted infrared radiation and encode temperature dependent signals. Each pixel represents a temperature value or a value proportional to heat emission, and any colors seen in the image come from a colormap applied to these underlying temperature readings.Count the number of humans in the image."

**Task T4 : Colorbar Identification**

- **Original**: "You are given a thermal image. Does it contain a color bar or temperature scale that maps colors to temperature values? Answer only with 'Yes' or 'No'."
- **Ablation**: "Thermal images encode temperature dependent signals. Each pixel represents a temperature value or a value proportional to heat emission, and any colors seen in the image come from a colormap applied to these underlying temperature readings. They may optionally include a color bar or temperature scale that visually maps colormap colors to corresponding temperature values. Such scales are typically placed along an edge of the image and indicate numeric temperature readings associated with the color gradient. Based on this context, does the given thermal image contain a color bar or temperature scale? Answer only with Yes or No."

**Task T4 : Colorbar Position Detection**

- **Original**: "You are given a thermal image. It contains a color bar or temperature scale that maps colors to temperature value. What is the location of the colorbar? Possible locations are top, left, bottom, right."
- **Ablation**: "Thermal images encode temperature dependent signals. Each pixel represents a temperature value or a value proportional to heat emission, and any colors seen in the image come from a colormap applied to these underlying temperature readings. They include a color bar or temperature scale that visually maps colormap colors to corresponding temperature values. Such scales are typically placed along an edge of the image and indicate numeric temperature readings associated with the color gradient. Based on this context, determine the location of the color bar in the given thermal image. Possible locations are top, left, bottom, or right."

**Task T4 : Colorbar Min/Max Extraction**

- **Original** : "You are given a thermal image with a color bar or temperature scale that maps colors to temperature value. What is the maximum temperature value in degree Celsius?"
- **Ablation**: "Thermal images encode temperature dependent signals. Each pixel represents a temperature value or a value proportional to heat emission, and any colors seen in the image come from a colormap applied to these underlying temperature readings. They include a color bar or temperature scale that visually maps colormap colors to corresponding temperature values. Such scales are typically placed along an edge of the image and indicate the temperature

range represented by the colormap. Using this definition, determine the maximum temperature value shown on the color bar in the given thermal image, expressed in degree Celsius."

- **Original** : "You are given a thermal image with a color bar or temperature scale that maps colors to temperature value. What is the minimum temperature value in degree Celsius?"
- **Ablation**: "Thermal images encode temperature dependent signals. Each pixel represents a temperature value or a value proportional to heat emission, and any colors seen in the image come from a colormap applied to these underlying temperature readings. They include a color bar or temperature scale that visually maps colormap colors to corresponding temperature values. Such scales are typically placed along an edge of the image and indicate the temperature range represented by the colormap. Using this definition, determine the minimum temperature value shown on the color bar in the given thermal image, expressed in degree Celsius."

**Task T5 : Thermal Reasoning**

- **Original** : "Given the thermal image with colourbar, determine whether the bodypart of the left person or the right person is hotter. Respond with only 'left' or 'right'."
- **Ablation**: "Thermal images encode temperature dependent signals. Each pixel represents a temperature value or a value proportional to heat emission, and any colors seen in the image come from a colormap applied to these underlying temperature readings. They include a color bar or temperature scale that visually maps colormap colors to corresponding temperature values. Such scales are typically placed along an edge of the image and indicate the temperature range represented by the colormap, enabling comparison of temperatures across different regions. Using this definition, determine which person's bodypart is hotter in the given thermal image. Respond only with left or right."

- **Original**: "Given the thermal image and the colourbar, rank the following body parts in order from highest to lowest temperature: chest, forehead and nose. List them from hottest to coolest."
- **Ablation**: "Thermal images encode temperature dependent signals. Each pixel represents a temperature value or a value proportional to heat emission, and any colors seen in the image come from a colormap applied to these underlying temperature readings. They include a color bar or temperature scale that visually maps colormap colors to corresponding temperature values. Such scales are typically placed along an edge of the image and indicate the temperature range represented by the colormap, enabling comparison of temperatures across different regions. Using this definition, rank the chest, forehead and nose in the given thermal image from highest to lowest temperature. List them from hottest to coolest."

**Task T6 : Temperature Estimation**

- **Original:** "Given the thermal image, what is the temperature at the coordinates (x,y)? The temperature scale is in degrees Celsius. Return a single numerical value rounded to one decimal place (e.g., 17.6)."
- **Ablation:** "Thermal images encode temperature dependent infrared signals. Each pixel represents a temperature value or a value proportional to emitted heat, and any visible colors come from a colormap applied to these values. A visible color bar or temperature scale maps colormap colors to numeric temperature readings in degrees Celsius. Given image pixel coordinates (x,y) with origin at the top-left, x increasing to the right and y increasing downward, report the temperature at the specified coordinates as a single numeric value in degrees Celsius rounded to one decimal place (for example, 17.6)."

### B.7.2    KEY FINDINGS

Our ablation reveals three clear trends across models and tasks.

- Models with reasonable visual grounding (InternVL-14B, MiniCPM, Qwen-VL-2.5) show large gains for simple tasks when contextual modality descriptions are added.
    - Qwen-VL-2.5 — T1 FLIR: $0.71 \rightarrow 0.96$ and LLVIP: $0.72 \rightarrow 0.96$
    - Qwen-VL-2.5 — T2 FLIR: $0.61 \rightarrow 0.98$ and LLVIP: $0.80 \rightarrow 0.99$
    - InternVL-14B — T2 FLIR: $0.86 \rightarrow 0.99$ and LLVIP: $0.97 \rightarrow 1.00$

Table 12: Performance of models before and after prompt ablations. ↓ indicates lower mae is better, and ↑ indicates higher accuracy is better.

| Task | Dataset | InternVL (Original) | InternVL (Ablation) | MiniCPM (Original) | MiniCPM (Ablation) | Qwen2.5-VL (Original) | Qwen2.5-VL (Ablation) | BLIP-2 (Original) | BLIP-2 (Ablation) |
|---|---|---|---|---|---|---|---|---|---|
| T1 | FLIR (↑) | 0.96 | 0.97 | 0.95 | 0.96 | 0.71 | 0.96 | 0.46 | 0.34 |
|  | LLVIP (↑) | 1.00 | 1.00 | 0.98 | 0.99 | 0.72 | 0.96 | 0.22 | 0.43 |
| T2 | FLIR (↑) | 0.86 | 0.99 | 0.91 | 0.89 | 0.61 | 0.98 | 0.77 | 0.67 |
|  | LLVIP (↑) | 0.97 | 1.00 | 0.93 | 0.95 | 0.80 | 0.99 | 0.77 | 0.77 |
| T3 | FLIR (↓) | 2.70 | 2.53 | 3.70 | 2.90 | 3.78 | 3.20 | 4.69 | 4.65 |
|  | LLVIP (↓) | 0.73 | 0.60 | 1.09 | 0.80 | 1.09 | 0.88 | 2.99 | 2.94 |
| T4 | Detect (↑) | 1.00 | 1.00 | 1.00 | 1.00 | 1.00 | 1.00 | 0.50 | 0.50 |
|  | Position (↑) | 1.00 | 1.00 | 0.99 | 0.99 | 0.99 | 0.97 | 0.25 | 0.25 |
|  | Min (↓) | 0.00 | 0.00 | 0.00 | 0.00 | 2.66 | 1.36 | 42.58 | 25.62 |
|  | Max (↓) | 0.00 | 0.00 | 0.00 | 0.00 | 0.00 | 0.00 | 209.80 | 20.80 |
| T5 | Single (↑) | 0.32 | 0.20 | 0.27 | 0.26 | 0.42 | 0.39 | 0.16 | 0.42 |
|  | Double (↑) | 0.51 | 0.50 | 0.40 | 0.41 | 0.41 | 0.51 | 0.39 | 0.39 |
| T6 | Arrow (↓) | 5.28 | 4.30 | 6.32 | 5.95 | 4.75 | 4.46 | 12.74 | 12.77 |
|  | Coordinate (↓) | 3.48 | 4.12 | 4.00 | 4.85 | 3.65 | 4.22 | 13.08 | 12.74 |
|  | Region (↓) | 2.18 | 2.51 | 4.23 | 3.29 | 2.91 | 3.22 | 14.73 | 14.73 |
| T7 | 2ft (↓) | 1.01 | 2.04 | 2.15 | 2.45 | 1.05 | 1.63 | 16.96 | 16.96 |
|  | 6ft (↓) | 1.12 | 2.26 | 2.02 | 2.47 | 1.00 | 1.28 | 16.35 | 16.35 |
|  | 10ft (↓) | 1.70 | 2.53 | 1.85 | 2.01 | 1.00 | 1.23 | 15.43 | 15.43 |

**Interpretation:** Architecture is not the bottleneck; a short textual description helps models *anchor* the visual signal and correctly name the modality.

- Across temperature-comparison and temperature-estimation tasks, gains are small, inconsistent, or negative.
  - Qwen T6 Arrow: 4.75 → 4.46 (small improvement)
  - MiniCPM T6 Coordinate: 4.00 → 4.85 (worse)
  - InternVL T7 (2ft): 1.01 → 2.04 (worse)

**Interpretation:** Thermal-physics descriptions in prompts cannot replace the sensor-level priors needed to map pixel or colormap values to temperature. The underlying challenge is the lack of thermal-domain grounding in model pretraining.

- BLIP-2 often degrades when given extra context:
  - T1 FLIR: 0.46 → 0.34
  - T2 FLIR: 0.77 → 0.67
  - T6/T7: unchanged and poor

**Interpretation:** When a model lacks basic thermal–visual alignment, prompt engineering cannot compensate for that gap. This aligns with the reviewer intuition that prompting alone is insufficient for reliable thermal reasoning.

# C VISUALIZATIONS

## C.1 TEMPERATURE DISTRIBUTION OF THERMEVAL-D DATASET

Outdoor data were collected during summer evenings, with ambient temperatures ranging from 30°C to 37°C. In contrast, indoor environments were air-cooled and maintained within a temperature range of 25–29°C.

| Metric | Mean | Median | Std. Dev. | Max | Min |
|---|---|---|---|---|---|
| **Max Temperature** | 36.56 | 35.80 | 3.48 | 52.50 | 30.90 |
| **Min Temperature** | 24.62 | 24.85 | 4.09 | 31.40 | 2.90 |
| **Temperature Range** | 11.94 | 11.60 | 4.48 | 32.10 | 4.00 |

Table 13: Summary statistics of temperature measurements from the thermal images. The *maximum temperature* refers to the highest recorded value in an image, while the *minimum temperature* corresponds to the lowest. The *temperature range* is computed as the difference between the maximum and minimum temperatures. The table presents various statistical measures across the dataset.

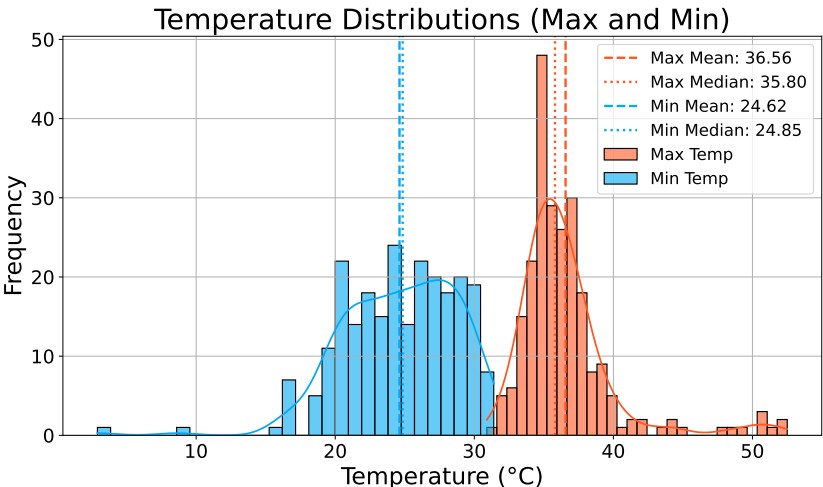

Figure 6: Histogram showing the distribution of minimum and maximum temperature values across all thermal images. Minimum temperatures are predominantly in the range of 20–30°C, while maximum temperatures typically fall within 30–40°C.

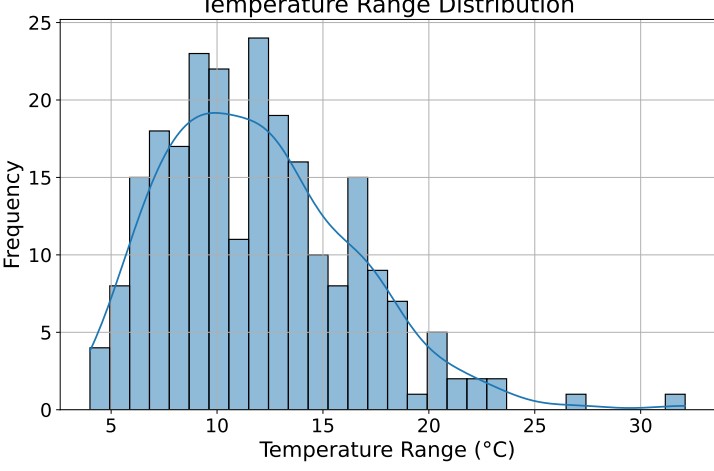

Figure 7: Histogram illustrating the distribution of temperature ranges (maximum minus minimum) across all thermal images. The majority of images exhibit a temperature range between 5–10°C.

