# OpenReview forum: "ThermEval: A Structured Benchmark for Zero-Shot Evaluation of Vision-Language Models on Thermal Imagery"
_ICLR.cc/2026/Conference — ICLR 2026 Conference Desk Rejected Submission_

### Official Review · Reviewer_Ath4 · 2025-10-29

**Soundness:** 2
**Presentation:** 2
**Contribution:** 1
**Rating:** 2
**Confidence:** 4

**Summary:**

The authors present a thermal image understanding benchmark for VLMs. It includes 500 image-QA pairs where all images have pixel-level annotations and questions from templates. The authors then do zero-shot evaluation for multiple open-source VLMs. The results show that most VLMs can distinguish thermal images from RGB images but still suffer from color map understanding.

**Strengths:**

1. The authors present a new thermal evaluation dataset with precise annotation.
2. The paper provides an initial study on zero-shot VLM capabilities in the thermal image domain.

**Weaknesses:**

1. Contribution In terms of ML/CV: While the benchmark can be important for a specific domain, is it important for the VLM field as a whole? What's the difference between lots of chart understanding / depth map understanding compared to the thermal images? The reviewer feels like it's a subdomain of the chart understanding and not as important/popular than benchmarking models' understanding on other modality inputs like optical flow or depth map from the computer vision perspective. As it's an machine learning conference, it would be great if the authors can address the problem from this perspective more.

2. Experimental Design: The goal of the paper is to analyze VLM's capability on thermal images and provide valuable insights for practitioners and researchers to move forward; however, the reviewer believes that the analyses are just surface-level due to the insufficient depth of experimental design. For all tasks, the authors assume that VLMs already understand the definition of thermal images and thus did not provide clear definition on it but the reviewer suggests that putting the definition of thermal images can be a great ablation studies. For instance, in task 1, the authors found that BLIP-2 and PaliGemma-2 perform bad when fed with a simple prompt "Is this a thermal image or an RGB image" (in L177, Table 1). However, it's unclear if it's because these models just did not have the capability to understand this kind of modality or it's because the instruction is not clear enough. What if the prompts be "The thermal images is captured.... where the resulting images are usually colored in xxx color map; the RGB images on the other hand... Given this definition, is the following image a thermal image or an RGB image?" From a user perspective it does not make sense for us not to do some simple prompt tweaking and come up with a conclusion directly. For this task, the author did not test any adversarial RGB examples as test cases as well. For task 3 and beyond, when observing the model performing bad on such questions, have the authors tried to provided few-shot prompting or deeper analyses on why it didn't work? Finally, from a practical perspective, do we usually need to understand a scene from only thermal images or should be thermal plus RGB images? The authors can discuss more about the connection between these benchmark questions and the real-world application as well, especially when the questions are all easy and created from a template.

3. Evaluation Protocol: The authors decided to use LLM-as-a-judge because the outputs often vary structurally (L314), resulting in 97% consistent compared to the human evaluation (L348). However, multiple libraries, such as VLLM, have support structure output for open-source VLMs and the authors can try to use that to make it better when it's a easy regression or yes/no question. For the results, it would be much better if the authors can do bootstrapping and report mean and standard deviation / confidence interval to ensure the reliability. Also, the authors did not run any proprietary models, lacking important baselines to compare.

**Questions:**

Please read the weakness section.

---

> ### Author Response · Authors · 2025-11-17
> **Thermal modality is important for VLMs.**
>
> 1. Contribution In terms of ML/CV.........if the authors can address the problem from this perspective more
>
> We thank the reviewer for their thoughtful comment and address their concern below
>
> ## **Thermal is a Distinct Modality**
> Thermal sensing captures emitted infrared radiation that varies with temperature. This introduces physical interpretation demands that do not arise in RGB, depth, or chart inputs.
> * Chart inputs contain explicit structure such as axes, legends, and symbolic markers that guide visual parsing. Thermal images contain none of these cues, and the model must infer structure from raw temperature fields.
> * RGB reflects visible light and depth encodes geometric distance, while thermal directly represents temperature through pixel intensities. Interpreting these intensities requires connecting image appearance to physical properties and reasoning over spatial temperature gradients.
> * Thermal images lack human designed boundaries or labels, so the model must discover semantic regions, differentiate people from background, and infer temperature driven patterns without guidance.
>
> ## **Real World Impact of Thermal Imagery**
> Thermal sensing is used extensively in settings where RGB vision is unreliable.
> * **Healthcare**: Non contact fever screening and physiological monitoring including facial thermal analysis , sleep monitoring and even including automatic apnea severity and type classification [1][2][3][4][5].
> * **Crowd management and rescue**: Thermal sensing offers robust visibility in smoke and fog which improves crowd and aerial monitoring in difficult conditions [6][7][8].
> * **Industrial and infrastructure safety**: It supports early detection of overheating equipment and invisible gas leaks, which reduces failure risk in critical systems [9].
> * **Autonomous navigation and surveillance**: It enhances perception under low light and adverse weather [10].
>
> Existing VLM evaluations on RGB thermal content reveal large performance gaps, indicating reliance on RGB specific cues rather than true multimodal generalization [11]. Our benchmark further shows that models that perform well on RGB tasks still fail on basic thermal concepts, often defaulting to fixed body temperature priors and treating colormapped thermal images as RGB photographs.
>
> ## **Advancing the field : Evaluating general multimodal intelligence.**
> * ThermEval provides a first evaluation suite dedicated to thermal reasoning and reveals consistent weaknesses across models.
> * Models default to fixed temperature priors, misinterpret colormap representations, and show difficulty in localizing semantic regions. These patterns occur across architectures, indicating a broader challenge in multimodal reasoning.
> * ThermEval complements existing RGB, chart, and depth benchmarks by introducing a capability that has not been evaluated before: interpreting a physical signal rather than a symbolic or visually structured one.
> * Thermal reasoning highlights missing abilities in current VLMs:
>     * not grounding pixel values to physical quantities
>     * unable to  suppress language priors, as noted in our paper(L451)
>     * not interpreting physical signals rather than relying on semantic pattern matching
> * ThermEval consolidates three complementary evaluation types within one framework:
>     * classification of thermal versus RGB,
>     * regression for physical temperature estimation,
>     * reasoning over spatial temperature patterns.
>
> ## **References**
> [1] Huang et al., 2021. Thermography based measurement of breathing during sleep. CVPRW. https://doi.org/10.1109/CVPRW53098.2021.00430
>
> [2] Kwasniewska et al., 2021. Respiratory rate estimation with transformers and CNNs. CVPRW. https://doi.org/10.1109/CVPRW53098.2021.00427
>
> [3] Ashrafi et al., 2022. Charlotte-ThermalFace annotated thermal face dataset. Infrared Physics & Technology. https://doi.org/10.1016/j.infrared.2022.104209
>
> [4] Akbarian et al., 2020. Distinguishing obstructive vs. central apneas in infrared sleep video. JMIR. https://doi.org/10.2196/17252
>
> [5] Akbarian et al., 2021. Noncontact sleep apnea severity estimation using infrared video. JMIR. https://doi.org/10.2196/26524
>
> [6] Liu et al., 2021. Cross modal learning for RGBT crowd counting. CVPR. https://arxiv.org/abs/2012.04529
>
> [7] Peng et al., 2020. RGB-T Crowd Counting from Drone. ACCV. https://dl.acm.org/doi/abs/10.1007/978-3-030-69544-6_30
>
> [8] Kniaz et al., 2018. ThermalGAN: Multimodal Color-to-Thermal Image Translation for Person Re-Identification in Multispectral. ECCV Workshops. https://link.springer.com/chapter/10.1007/978-3-030-11024-6_46
>
> [9] Wang et al., 2024. Invisible gas detection with RGB thermal cross attention. CVIU. https://arxiv.org/abs/2403.17712
>
> [10] Jia et al., 2021. LLVIP visible infrared paired dataset for low light vision. ICCV. https://bupt-ai-cz.github.io/LLVIP/
>
> [11] Moshtaghi et al., 2025. RGB Th Bench for thermal understanding in VLMs. arXiv. https://doi.org/10.48550/arXiv.2503.19654

---

> > ### Author Response · Authors · 2025-11-19
> > **Evaluating proprietary models, larger open-source models and fine-tuning**
> >
> > We thank the reviewers for their constructive feedback. All three reviewers requested evaluations with: i) proprietary, ii) larger open-source, and iii) finetuned models. We have incorporated these additions and report the results below.
> >
> > ## **Proprietary and Large Open Source Models**
> >
> > We have extended our analysis to include some proprietary and large open source models,
> > * GPT 4o,
> > * Gemini 2.5 Flash,
> > * Claude Haiku,
> > * Qwen A22, a large open source model with 235B parameters.
> >
> > These models were evaluated on Tasks T5 (thermal reasoning), T6 (temperature estimation), and T7 (temperature estimation at varying depth). We focused on these tasks because they are the most challenging. Some open source models already perform near perfectly on the remaining tasks, leaving little room for meaningful comparison.
> >
> > We report the result in the table below.
> >
> > ### **Comparison of VLM performance on Tasks 5, 6, and 7. x means the model did not answer, ? means the model size is unknown, ↓ means lower is better, and ↑ means higher is better.**
> > |Model|Params|T5 Double (Acc ↑)|T5 Single (Acc ↑)|T6 Coords (MAE ↓)|T6 Arrow (MAE ↓)|T6 Region (MAE ↓)|T7 2ft (MAE ↓)|T7 6ft (MAE ↓)|T7 10ft (MAE ↓)|
> > |-|-|-|-|-|-|-|--|-|-|
> > |**Qwen A22**|235B|0.58|0.27|3.96|4.21|3.01|1.97|2.09|2.24|
> > |**Gemini 2.5 Flash**|?|0.54|0.28|3.81|**3.48**|2.50|1.30|1.80|1.96|
> > |**GPT 4o**|?|0.46|0.34|x|x|x|x|x|x|
> > |**Claude Haiku 4.5**|?|0.28|**0.60**|4.28|4.45|2.47|1.37|1.57|1.90|
> > |**Best open source Performance**|max 38B|**0.61**|0.42|**3.48**|4.21|**1.76**|**1.01**|**1.00**|**1.00**|
> > |**Human Performance**|--|0.84|0.54|--|2.73|2.04|1.23|1.20|1.22|
> >
> >
> > ### **Findings**
> > * Larger open source and proprietary models still struggle with thermal reasoning and the performance gap between humans and VLMs persists.
> > * Increasing model size does not improve performance on thermal tasks, and larger models (Qwen A22 235B) show the same qualitative limitations as smaller ones (max 38B).
> > * GPT 4o declines to provide temperature estimates and returns responses such as “I cannot determine the exact temperature” or “I am unable to provide an exact temperature estimate”. We view this as an appropriate safety guardrail because a reliable VLM should avoid confident answers when uncertain.
> > * On MCQ based (Task-5 ordering body parts) tasks where GPT 4o does respond, its performance is comparable to other open source models and remains below human accuracy.
> >
> > ---
> >
> > ## **Finetuning Experiments**
> > We fine tuned Qwen 2.5 VL with 7B parameters on the ThermEval dataset using LoRA adapters to illustrate the dataset’s utility. The setup and results are summarised below.
> >
> > ### **Comparison of finetuned Qwen VL 2.5 (7B) with human performance and all other models. ↓ means lower is better, ↑ means higher is better, and - indicates not applicable**.
> > |Task|Best Model (Zeroshot)|Human Performance|Qwen 2.5 7B Zeroshot|Qwen 2.5 7B Finetuned| Qwen Δ (Finetuned − Zeroshot)|
> > |-|-|-|-|-|-|
> > |**T1 FLIR (Acc ↑)**|**1.00**|0.97|0.71|**1.00**|+0.29|
> > |**T1 LLVIP (Acc ↑)**|**1.00**|0.98|0.71|**1.00**|+0.29|
> > |**T2 FLIR (Acc ↑)**|**1.00**|0.98|0.61|**1.00**|+0.39|
> > |**T2 LLVIP (Acc ↑)**|**1.00**|0.98|0.80|**1.00**|+0.20|
> > |**T3 FLIR (MAE ↓)**|2.72|**1.73**|3.78|1.85|−1.93|
> > |**T3 LLVIP (MAE ↓)**|0.51|**0.30**|1.09|0.55|−0.54|
> > |**T4 Detect (Acc ↑)**|1.00|**1.00**|1.00|**1.00**|0.00|
> > |**T4 Position (Acc ↑)**|1.00|**1.00**|0.99|**1.00**|+0.01|
> > |**T4 Max (MAE ↓)**|0.00|**0.00**|0.00|**0.00**|0.00|
> > |**T4 Min (MAE ↓)**|0.00|**0.00**|2.66|**0.00**|−2.66|
> > |**T5 Double (Acc ↑)**|0.61|**0.84**|0.41|0.58|+0.17|
> > |**T5 Single (Acc ↑)**|**0.60**|0.54|0.42|0.56|+0.14|
> > |**T6 Cords (MAE ↓)**|3.48|--|3.65|**1.58**|−2.07|
> > |**T6 Arrow (MAE ↓)**|3.48|2.73|4.75|**1.55**|−3.20|
> > |**T6 Region (MAE ↓)**|1.76|2.04|2.91|**1.03**|−1.88|
> > |**T7 2ft (MAE ↓)**|1.01|1.23|1.05|**0.53**|−0.52|
> > |**T7 6ft (MAE ↓)**|1.00|1.20|1.00|**0.49**|−0.51|
> > |**T7 10ft (MAE ↓)**|1.00|1.22|1.00|**0.61**|−0.39|
> >
> > ### **Findings**
> > * Finetuning Qwen VL 2.5 allows it to outperform the much larger Qwen A22 235B and all evaluated closed source as well as open source models.
> > * The finetuned model matches or exceeds human performance on most tasks.
> > * These results show that ThermEval delivers meaningful supervision for thermal reasoning and establishes a reliable benchmark for advancing thermal modality capabilities in VLMs.
> >
> > ### **Experimental Setup**
> > * We fine tuned Qwen2.5 VL 7B Instruct for 5 epochs using LoRA with rank (r = 16) and scaling factor (alpha = 16), applied to the query, key, value, and output projection layers with dropout (0.1). Training used Paged AdamW 32 bit with a fixed learning rate (5 x 10^-6), no warmup, batch size 4 per device.
> > * The dataset was split into three stratified folds to balance tasks and subtasks, ensuring each VQA sample was seen exactly once and enabling full dataset evaluation without repetition.

---

> > > ### Comment · Reviewer_Ath4 · 2025-11-21
> > > **Reviewer's Response**
> > >
> > > Thanks for the additional experiments but the reviewer's concern remains there.
> > >
> > > First, the authors did not test if prompting with additional text helps to the model solve the problem in a zero-shot manner. Second, the above results show that the problem can be easily resolved by SFT on small number of data in this domain (i.e., matching or exceeding human performance), which again echos my point in the initial rebuttal: what's the insight of the proposed 6 tasks if they didn't guide what future work should work on? Finally, the authors did not test models on adversarial images. If the authors really want to differentiate the thermal domain from other charts, I feel like this is a necessary task.
> > >
> > > Again, I appreciate the paper's contribution of releasing a large-scale, precise thermal image dataset, which can be good resources for future model training. However, I still feel that the narration seems to be too narrow and the proposed benchmark does not seem to be insightful enough to inspire future work.
> > >
> > > I might reconsider the score after reading other reviewer's response but now I lean towards maintaining my score. Feel free to   discuss further and clarify these concerns. Thanks!

---

> > ### Comment · Reviewer_Ath4 · 2025-11-21
> > **Reviewer's Response**
> >
> > Thanks for the rapid reply and I completely agree that thermal images can be useful for multiple downstream applications in our everyday lives. However, the following sentence in the rebuttal does not seem to be convincing to me: "Chart inputs contain explicit structure such as axes, legends, and symbolic markers that guide visual parsing. Thermal images contain none of these cues, and the model must infer structure from raw temperature fields."
> >
> > The illustration in Figure 1 shows that thermal image can still be paired with legends and thus I'm not sure the actual difference between that and the conventional chart understanding task. I mean it's useful to provide, and open-source this thermal image dataset such that all future models can train or evaluate on this dataset. Yet, the reviewer is not convinced that there's any fundamental difference compared to the chart domain.

---

> > > ### Comment · Reviewer_c8GH · 2025-11-23
> > > **Discussion from Another Reviewer**
> > >
> > > I think the reviewer Ath4's concerns are interesting, and I also want to participate in the discussion.
> > >
> > > I understand the connection between thermal imaging images and chart images mentioned by reviewer Ath4. As I said in my comment, I think thermal imaging is a kind of fake color image, which is an artificial data expression, not the so-called modality. However, thermal imaging maps also seem to have a certain degree of irreplaceability. For example, the fine-tuned model in chart data or depth map data that does not contain thermal imaging maps obviously cannot understand thermal imaging maps well. It is indeed a completely different form of information expression, and it is not difficult for a visual system to understand. I am very concerned about the discussion about this, because all kinds of fake color images (such as PCA representation of hyper-spectral images, attention heat map, segmentation visualization, etc.) have the meaning of being made into benchmarks. I look forward to including such content, but I think a comprehensive benchmark that discusses all kinds of fake colors is more sufficient.

---

> > > > ### Comment · Reviewer_c8GH · 2025-11-23
> > > > **Discussion from Another Reviewer**
> > > >
> > > > In addition, i'm still not sure about whether this work is suitable as an ICLR submission. It seems that everyone is more concerned about its contribution to the field of machine learning and technical understanding. We are not experts in the field of thermal imaging. We can't really perceive whether the benchmark matches the thermal imaging in practical application. This is also the reason why I give 3 for confidence.

---

> > > > ### Comment · Reviewer_f9Nx · 2025-11-27
> > > > **Discussion from Reviewer f9Nx**
> > > >
> > > > I think the discussion on thermal image representation is quite interesting. As reviewer c8GH noted, a thermal image uses a man-made color map for visualization, so you can apply any color map as long as it stays perceptually uniform. My concern is that training on only one specific mapping, even though there is a common mapping for thermal image, might reduce the model’s usefulness and generality.
> > > >
> > > > So actually the real question is whether this approach makes sense at all. We first turn the raw signal into a color image for visualization, then ask a VLM to learn from that. This visualization step is only for people, not for machines. If a VLM cannot handle it in a zero-shot way and still needs finetuning, then it may be more reasonable to train a model that works on the raw signal directly, before any visualization.

---

> > > > > ### Comment · Reviewer_f9Nx · 2025-11-27
> > > > >
> > > > > And if this does make sense, since a VLM with pretraining can infer semantics and structure, which might help these tasks. What's the best visualization method for that? Is the colormap choice would influence performance. And even more shall we extend this method to all the other modality? (e.g. depth, normal map, segmentation) by first visualize them and let VLM to infer some meaningful information?

---

> > > > > > ### Author Response · Authors · 2025-11-29
> > > > > > **False Colormapped Thermal Imagery Is the Standard Interface for Thermal Modality**
> > > > > >
> > > > > > Thank you for raising these points. We summarize and respond to your core concerns here and in the following messages.
> > > > > >
> > > > > > ---
> > > > > > ## **Visualized Physical Modalities Are Standard and Effective for VLM Reasoning**
> > > > > >
> > > > > > > “The scope feels narrow..current VLMs have never seen this modality, so the results feel expected.”
> > > > > >
> > > > > > > “Is it even sensible to let a VLM analyze thermal data? Machines could read raw radiance directly rather than looking at the rendered image.”
> > > > > >
> > > > > > > “Should we instead visualize every modality (depth, normals, segmentation) and let VLMs reason?”
> > > > > >
> > > > > > We appreciate the reviewer’s concern. Our approach follows a standard practice in VLM research, where physical measurements are converted into visually interpretable representations. Many recent works show that VLMs learn reliably from false colormapped physical modalities and that this is the usual operable form for depth, thermal, and other sensor data. **VLMs are not limited to RGB, only predominantly trained on it.**
> > > > > >
> > > > > > * **SpatialBot (ICRA 2025)**[1]: Uses depth rendered as false colormaps for spatial reasoning, motivated by the same observation that VLMs excel with RGB visual inputs but need visualized physical cues for modality specific reasoning.
> > > > > > * **Leveraging MLLMs for Thermal Perception in Autonomous Driving (MDPI 2024)**[2]: Shows that GPT and Gemini can interpret thermal images directly, confirming that thermal colormaps support high level understanding tasks.
> > > > > > * **IRGPT (ICCV 2025)**[3]: Uses false-color thermal images primarily for detection and scene interpretation, but does not address semantic to physical reasoning (temperature estimation, ranking, or interpreting colorbars). ThermEval therefore targets a different capability class not covered in prior work.
> > > > > > * **Thermal Image Driven Clothing Insulation Estimation (Energy and Buildings 2025)** [4]: Uses false colormapped thermal images for quantitative prediction of clothing insulation, reinforcing that thermal visualizations are a standard modality for extracting physically meaningful quantities.
> > > > > > * **Zero Shot Anomaly Detection in Battery Thermal Images (EUSIPCO 2025)** [5]: Applies pretrained VQA models to false colormapped thermal images for battery anomaly detection, showing that text guided reasoning over thermal colormaps can support diagnostic tasks without modality specific training.
> > > > > >
> > > > > > Together, these works show that false color thermal images are the standard operable form of the modality and are widely used across detection, reasoning, and prediction tasks in foundational and applied settings.
> > > > > >
> > > > > > Below are representative works where models predict physical quantities from visual inputs, aligning with our temperature estimation task
> > > > > > * DepthLM (Meta) [6]: Predicts metric depth from RGB with light supervised finetuning and reaches accuracy comparable to specialized depth models, showing that VLMs achieve expert level performance without architectural or loss changes.
> > > > > > * Depth and Height Perception in Large VLMs (CVPRW 2025) [7]: Demonstrates that VLMs infer 3D structure and height from visual encodings rather than raw physical measurements, emphasizing that visualization is the mechanism enabling geometric reasoning.
> > > > > >
> > > > > > These efforts show that predicting physical quantities from image inputs is well established and effective. They support three conclusions
> > > > > > 1. VLMs operate on visual representations, not raw signals.
> > > > > > 2. Visualized physical modalities are valid and widely used inputs.
> > > > > > 3. Small supervised finetuning is sufficient to unlock physical grounding.
> > > > > >
> > > > > > Our paper follows the same principle. Once thermal signals are rendered visually, they form a modality that VLMs can naturally learn from, mirroring patterns seen in depth and other physical signals.
> > > > > >
> > > > > > Regarding the question,
> > > > > > >“If VLMs need SFT to handle this, is this approach meaningful?”
> > > > > >
> > > > > > DepthLM [6] states: “SFT with sparse labels is sufficient to unlock strong 3D understanding, no architectural change is needed.” Our findings align with this line of work. VLMs already have the representational capacity, and the main missing component is exposure to the modality. We show that thermal signals, when visualized in a principled manner, follow the same learning pattern observed for depth and other physical modalities.
> > > > > >
> > > > > > ---
> > > > > > [1] SpatialBot: Precise Spatial Understanding with VLMs , ICRA'25 https://ieeexplore.ieee.org/document/11128671
> > > > > >
> > > > > > [2] Leveraging MLLMs for Thermal Perception in Autonomous Driving, MDPI'24  https://www.mdpi.com/2673-4052/5/4/29
> > > > > >
> > > > > > [3] IRGPT:  ICCV'25 https://arxiv.org/abs/2507.14449
> > > > > >
> > > > > >
> > > > > > [4] Thermal Image Driven Clothing Insulation Estimation, Energy and Buildings 2025. https://doi.org/10.1016/j.enbuild.2025.116720
> > > > > >
> > > > > > [5] Zero Shot Anomaly Detection in Battery using Thermal Images, EUSIPCO'25 https://eusipco2025.org/wp-content/uploads/pdfs/0000711.pdf
> > > > > >
> > > > > > [6] DepthLM : Metric Depth From Vision-Language Models” https://arxiv.org/pdf/2509.25413
> > > > > >
> > > > > > [7] Understanding Perception in Large Visual-Language.  CVPR'25 https://arxiv.org/abs/2408.11748

---

> > > > > > > ### Author Response · Authors · 2025-11-29
> > > > > > > **Raw Radiance Is Not the Standard Input for Downstream Tasks**
> > > > > > >
> > > > > > > ## **Thermal images and NOT raw radiance are the real inputs used in practice**
> > > > > > >
> > > > > > > > (2) “Is it even sensible to let a VLM analyze thermal data? Machines could read raw radiance directly rather than looking at the rendered image.”
> > > > > > >
> > > > > > > A key practical point is that raw radiance is often inaccessible outside the manufacturer’s pipeline, just as RGB cameras do not expose raw illuminance. Most thermal cameras such as FLIR, Seek, HikVision, and DJI IR provide only processed thermal images. Likewise, all major public datasets (FLIR ADAS [1], LLVIP [2], KAIST [3], OpenThermalPose [4]) release false color thermal imagery rather than radiometric streams. ThermEval additionally provides the underlying temperature field, enabling proper evaluation of physically grounded reasoning rather than only semantic tasks
> > > > > > >
> > > > > > > In practice, false color thermal images are what downstream systems and analytics pipelines use. Evaluating VLMs on these images is therefore the realistic operational setting for thermal perception.
> > > > > > >
> > > > > > > ## **Additional insights from SAM 3 (Segment Anything Model 3)**
> > > > > > >
> > > > > > > SAM3 [5] independently confirms that thermal images are already treated as a meaningful non-RGB modality in current foundation-model evaluation.
> > > > > > >
> > > > > > > The SAM3 appendix reports:
> > > > > > >
> > > > > > > >“SAM 3 struggles to generalize to fine-grained out-of-domain concepts, particularly in niche visual domains such as **thermal** imagery, in a zero-shot manner.”
> > > > > > >
> > > > > > > Furthermore, the ODinW13 benchmark used by SAM3 includes ThermalDP, a thermal detection dataset. SAM3 shows the same behaviour that we observe:
> > > > > > > - zero-shot performance on thermal is poor
> > > > > > > - few-shot fine-tuning quickly adapts the model
> > > > > > >
> > > > > > > This mirrors our finding that the issue is not model capacity but missing modality grounding. Thus, evaluating VLMs on visualized thermal data is fully aligned with, and supported by, contemporary VLM literature.
> > > > > > >
> > > > > > > ## **Why not train a raw-thermal specialist model instead?**
> > > > > > >
> > > > > > > Raw signal thermal models exist for tasks such as super resolution, but they address different problems. Our benchmark focuses on VLM style semantic and linguistic reasoning, which raw thermal networks cannot perform.
> > > > > > >
> > > > > > > Raw thermal models cannot
> > > > > > > * reason over semantic regions such as forehead versus chest,
> > > > > > > * answer language conditioned queries,
> > > > > > > * handle multi step comparisons such as which person is hotter.
> > > > > > >
> > > > > > > These abilities match real VLM use cases, for example
> > > > > > > * What is the temperature of this marked location,
> > > > > > > * Estimate the fever from the face region,
> > > > > > > * Compare the temperature between two objects,
> > > > > > >
> > > > > > > Such queries require semantic grounding, language understanding, and physical reasoning that raw signal thermal networks do not provide. Our benchmark therefore evaluates capabilities specific to VLMs rather than competing with raw thermal specialists.
> > > > > > >
> > > > > > > ## **Why SFT “working well” is not a limitation but a useful diagnostic**
> > > > > > > Our interpretation is that SFT performing well is itself informative. Recent work in related settings (DepthLM, SpatialBot) shows a similar pattern:
> > > > > > > - light supervised fine-tuning on visualized physical signals unlocks abilities that are already latent in VLMs. Our results suggest the same trend for thermal data.
> > > > > > > - The benchmark highlights that current VLMs are not missing model capacity, but rather exposure to this modality and the right grounding signal.
> > > > > > > - ThermEval helps identify where models struggle in zero-shot mode and what minimal supervision resolves.
> > > > > > >
> > > > > > > This is the type of diagnostic signal a benchmark aims to provide.
> > > > > > >
> > > > > > > ## **Summary**
> > > > > > > To summarize the argument:
> > > > > > > - Thermal images are not just for humans: They are the de facto operational interface for robotics, healthcare, security, and drones.
> > > > > > > - Visualized physical signals are already standard input to VLMs: Depth, segmentation,etc, none are provided as raw physical tensors in practice.
> > > > > > > - SFT success indicates missing modality grounding, not triviality: Mirrors results in VLM-depth, VLM-geometry, and VLM-spatial reasoning literature.
> > > > > > > - Specialist raw-signal models cannot replace VLMs: They lack semantic reasoning, language grounding, and generalization.
> > > > > > > - ThermEval fills a current gap by offering a structured way to evaluate how VLMs connect semantic cues with underlying physical information in a thermal modality.
> > > > > > >
> > > > > > > We hope this clarifies our position and demonstrates how ThermEval fits naturally into the current trajectory of VLM research.
> > > > > > >
> > > > > > > ---
> > > > > > >
> > > > > > > ## **References**
> > > > > > >
> > > > > > > [1] FLIR ADAS – FREE FLIR Thermal Dataset for Algorithm Training https://oem.flir.com/en-in/solutions/automotive/adas-dataset-form
> > > > > > >
> > > > > > > [2] LLVIP – A Visible-Infrared Paired Dataset for Low-Light Vision https://arxiv.org/pdf/2108.10831
> > > > > > >
> > > > > > > [3] KAIST Multispectral Pedestrian Dataset https://openaccess.thecvf.com/content_cvpr_2015/papers/Hwang_Multispectral_Pedestrian_Detection_2015_CVPR_paper.pdf
> > > > > > >
> > > > > > > [4] OpenThermalPose2: Extending the Open-Source Annotated Thermal Human Pose Dataset https://ieeexplore.ieee.org/document/11020744
> > > > > > >
> > > > > > > [5] SAM 3 https://arxiv.org/pdf/2511.16719

---

> ### Comment · Reviewer_Ath4 · 2025-11-23
> **Reviewer's Response**
>
> Thanks R#c8GH for joining this discussion thread. My expertise is centered around VLM eval/benchmarks rather than the thermal image domain. That's why my initial, direct reaction after reading this paper was, "oh, what's the difference between your task and other chart understanding ones," from the **model's perspective** rather than the downstream application.
>
> Similar to R#c8GH's status, while I'm not entirely sure how valuable the dataset is to the thermal image domain (I assume it has substantial contribution), I feel that the TA might not be general VLM practitioners or broader ML folks. (Also wanna let the AC knows my status such that we might have relative fair judgement at the end of the day.)
>
> I think it would be really helpful if the authors could reply to this thread to address a bunch of these concerns!

---

> ### Author Response · Authors · 2025-11-25
> **Thermal Images are not Charts.**
>
> We thank reviewers R#Ath4 and R#c8GH for their constructive comments and discussion as a response to our rebuttal. We summarize their key concerns below.
>
> **R#c8GH**
>
> (1) *“The fine tuned model in chart data...that does not contain thermal imaging maps obviously cannot understand thermal imaging maps well.”*
>
> (2) *“We are not experts in the field of thermal imaging. We cannot really perceive whether the benchmark matches the thermal imaging in practical application.”*
>
> (3) *"all kinds of fake color images ... have the meaning of being made into benchmarks. I look forward to including such content..."*
>
> **R#Ath4**
>
> (4) *“What is the difference between chart understanding or depth map understanding and thermal images?”*
>
> (5) *“What is the difference between your tasks and other chart understanding ones?”*
>
> (6) *“If SFT solves it easily, what is the insight?”*
>
> (7) *"what's the insight of the proposed 6 tasks"*
>
>
> We group concerns (1), (4) and (5) together and address them below. The remaining points are addressed in the next messages. We are also working other open points and will be sharing our result/analysis soon.
>
> ---
>
> To test the hypothesis (suggested by R#Ath4) that thermal images behave like charts, we evaluated ChartGemma [1], a state-of-the-art chart VQA model heavily trained on chart data. If thermal were equivalent to chart representations, ChartGemma should perform competitively on ThermEval. Its results show the opposite.
>
> ### **Table shows comparison of ChartGemma on ThermEval. ↓ means lower mae is better, and ↑ means higher accuracy is better.**
> |Metric|T1 FLIR ↑|T1 LLVIP ↑|T2 FLIR ↑|T2 LLVIP ↑|T3 FLIR ↓|T3 LLVIP ↓|T4 Detection ↑|T4 Position ↑|T4 Min ↓|T4 Max ↓|T5 Single ↑|T5 Double ↑|T6 Arrow ↓|T6 Coordinate ↓|T6 Region ↓| T7 2ft ↓|T7 6ft ↓|T7 10ft ↓|
> |-|-|-|-|-|-|-|-|-|-|-|-|-|-|-|-|-|-|-|
> |ChartGemma [1]|0.56|0.53|0.16|0.13|3.06|1.31|0.50|0.26|0|0|0.27|0.52|5.91|5.43| 5.43|4.44|3.56|3.25|
> |Best Model Zero Short Performance|1.00|1.00|1.00|1.00|2.72|0.51|1.00|1.00|0.00|0.00|0.61|0.60|3.48|3.48|1.76|1.01|1.00|1.00|
> |Fine-tuned Model Performance|1.00|1.00|1.00|1.00|1.85|0.55|1.00|1.00|0.00|0.01|0.56|0.58|1.55|1.58|1.03|0.53|0.49|0.61|
> |Random Chance|0.50|0.50|0.50|0.50|-|-|0.50|0.25|-|-|0.167|0.50|-|-|-|-|-|-|
> |Human Performance|0.97|0.98|0.98|0.99|1.73|0.30|1.00|1.00|0.00|0.00|0.84|0.54|2.73|-|2.04|1.23|1.20|1.22|
>
> [1] Masry, A. et all. ChartGemma: Visual instruction-tuning for chart reasoning in the wild. ACM ACL'25. https://aclanthology.org/2025.coling-industry.54.pdf
>
> ### **Insights**
> Despite excelling at chart QA, ChartGemma performs substantially worse than both open-source and proprietary VLMs on ThermEval. It systematically misinterprets thermal images:
> - It treats any colormap as implying a colorbar (high false positives in T4 detection).
> - When a colorbar exists, it incorrectly assumes a fixed position (always “top”).
> - It retains strong OCR ability but fails at basic modality recognition (T1–T2) ,human counting (T3), and eventually collapsing at all downstream reasoning (T5–T7).
>
> These results support the R#c8GH's intuition: chart-tuned models do not generalize to thermal imagery because the underlying representations differ fundamentally.
>
> ### **Confusion between Images and Charts**
>
> Thermal images and charts may appear similar because both can use colormaps, but they represent fundamentally different information.
>
> An RGB camera records an H×W×3 matrix of visible light intensities, while a thermal camera outputs an H×W×1 matrix of temperatures. Colormaps are applied only for visualization. The underlying data remain temperatures, not colors, and the colormap does not introduce axes, layout, or structured semantics. The colorbar simply displays the temperature scale.
>
> Any single channel data (depth, grayscale RGB, thermal) can be visualized using an arbitrary colormap. The appearance alone does not make such a visualization a chart. A colormap does not introduce axes, layout, or structured semantics. The colorbar simply indicates the temperature scale; it does not convert the underlying data into a chart.
>
> * Charts and heatmaps are human constructed artifacts. They are deliberately organized, cleaned, and structured to make the underlying information easy to interpret, with fixed spatial meaning, clear semantics, consistent grids, and predictable layouts.
> * Thermal images contain no designed structure; they are raw sensor measurements shaped by geometry, materials, emissivity, distance, and occlusion.
> * Even basic thermal tasks require multi step reasoning, for example detecting a person, localizing a region, interpreting the colorbar, and mapping pixel values to temperature.
> * Empirically, a SOTA Chart model (ChartGemma) performs worse than general purpose VLMs on our benchmark and shows systematic biases, indicating that thermal understanding requires interpreting a physical scene rather than reading a structured diagram.
>
> Therefore, we believe that thermal images are not charts.

---

> ### Author Response · Authors · 2025-11-25
> **Justification of the benchmark tasks and their relevance to practical applications**
>
> Reiterating the concerns for context
>
> (2) *“We are not experts in the field.....practical application.”*
>
> (7) *"what's the insight...tasks"*
>
> Building on the previous discussion, we outline the motivation for the benchmark tasks (7) and their connection to practical thermal imaging applications (2). We are working on the other open points and will share our analysis soon.
>
> Our benchmark reflects capabilities needed in real world thermal applications such as safety monitoring, detection in low visibility, industrial inspection, and non contact health assessment. They begin with basic abilities like identifying whether an image is thermal and gradually progress to higher level reasoning, such as estimating the temperature of a semantic region like the forehead. The objective is to assess fundamental skills that general purpose VLMs should reasonably possess.
>
> | Sno| Task|Difficulty| Description|Reason to Include|
> |-|-|-|-|-|
> | T1  | Modality Identification|Easy| Identify whether the image is thermal or not.| To assess a model’s basic ability to recognize the appearance of thermal images.|
> | T2  | Modality Identification under Colormap | Easy | Identify whether an image is thermal even when a different colormap is applied. | Different applications may require varying colormaps for improved visibility and interpretability.|
> | T3  | Human Counting| Moderate|Count the number of humans in the image.| Human localization in low visibility settings is a central application of thermal imaging.|
> | T4  | Colorbar Understanding| Moderate| Detect, localize, and extract the minimum and maximum values from a colorbar.| Without reading the colorbar, a model cannot estimate or reason about temperature reliably.|
> | T5  | Thermal Reasoning| Hard| Localize regions, interpret the colorbar, and compare temperatures to identify the warmer object. | Comparing object temperatures is a common use case for VLM-based thermal reasoning.|
> | T6–T7 | Temperature Estimation| Hard| Estimate temperature at a coordinate, arrow-marked location, or semantic region.| Models should estimate temperature in meaningful regions and must handle user-marked locations when vocabulary is insufficient.|
>
> ## **Insights**
> * Task T1, T2 and T4 are standalone basic steps for understanding thermal modality
> * Task T3, T5 , T6 and T7 are the tasks that forces the model to interpret thermal cues for answering queries.
>
> ## **Practical Usefulness of Benchmark Tasks**
>
> We design these tasks to reflect core applications of thermal imaging, noting that this list is non-exhaustive and additional use cases exist.
>
> ### **T3. Human Presence and Counting**
> Thermal person detection is essential in low-light and low-visibility environments.
>
> **Applications:** autonomous driving, surveillance, and occupancy monitoring.
>
> **Research examples:**
>
> * *KAIST Multispectral Pedestrian* (Hwang et al., CVPR 2015): widely used RGB–thermal for pedestrian detection and counting.
>   https://github.com/SoonminHwang/rgbt-ped-detection
>
> * *TODOS: Thermal sensOr Data-driven Occupancy Estimation System for Smart Buildings*, Rajabi et al., BuildSys 2023: uses a low-cost thermal sensor array and neural pipeline to estimate occupancy in commercial spaces.
>   https://dl.acm.org/doi/10.1145/3600100.3623753
>
> ### **T5. Thermal Reasoning (Relative Temperature Comparison)**
> Relative temperature patterns often provide more reliable cues than absolute values.
>
> **Applications:** fever and inflammation screening, occupational heat-stress monitoring, human–robot interaction, behavioral physiology.
>
> **Research examples:**
>
> * *Reading Between the Heat: Co-Teaching Body Thermal Signatures for Non-intrusive Stress Detection*, Xiao et al., IMWUT / Ubicomp: infers stress using relative temperature across facial and body regions.
>   https://dl.acm.org/doi/10.1145/3631441
>
> * *Comparative Study of Forehead and Core Temperature*, Chen et al., 2022: analyzes region-wise temperature differences for contactless health assessments.
>   https://pubmed.ncbi.nlm.nih.gov/36497956/
>
> ### **T6–T7. Temperature Estimation (Coordinate, Arrow, Region-based)**
> These tasks represent scenarios requiring numerical temperature extraction.
>
> **Applications:** industrial inspection, building diagnostics, medical screening, and robotic operation near heat sources.
>
> **Research examples:**
>
> * *Non-contact Infrared Body Temperature Assessment*, Foster et al., 2021: evaluates accuracy and limitations of infrared thermography for human temperature estimation.
>   https://www.ncbi.nlm.nih.gov/pmc/articles/PMC8328868/
>
> * *Infrared Thermography for Industrial Fault Detection*, Bagavathiappan et al., Infrared Physics: demonstrates pixel-level thermal analysis for hotspot detection in industrial systems.
>   https://doi.org/10.1016/j.infrared.2012.03.002
>
> In summary, these tasks highlight the fundamental reasoning steps needed for thermal interpretation and provide a clear framework for measuring how well current VLMs handle thermal Imagery.

---

> > ### Author Response · Authors · 2025-11-25
> > **Supervised Fine Tuning does not solve the problem with thermal modality.**
> >
> > Reiterating the reviewer concern:
> >
> > **R#Ath4** : (6) *“If SFT solves it easily, what is the insight?”*
> >
> > ---
> >
> > ## **Supervised Fine Tuning (SFT)**
> > R#Ath4 raises the concern that if small-scale SFT improves performance, the tasks may offer limited insight. From our perspective, the SFT results suggest that the main difficulty arises from limited exposure to the thermal modality, rather than from any fundamental shortcomings in the models.
> >
> > ### **Comparison of finetuned Qwen-VL 2.5 (7B) against zero-shot performance, human performance, and random chance. [Reshared for context]. ↓ means lower is better, and ↑ means higher is better.**
> > |Metric|T1 FLIR ↑|T1 LLVIP ↑|T2 FLIR ↑|T2 LLVIP ↑|T3 FLIR ↓|T3 LLVIP ↓|T4 Detection ↑|T4 Position ↑|T4 Min ↓|T4 Max ↓|T5 Single ↑|T5 Double ↑|T6 Arrow ↓|T6 Coordinate ↓|T6 Region ↓| T7 2ft ↓|T7 6ft ↓|T7 10ft ↓|
> > |-|-|-|-|-|-|-|-|-|-|-|-|-|-|-|-|-|-|-|
> > |**Best Model Zero Shot Performance**|1.00|1.00|1.00|1.00|2.72|0.51|1.00|1.00|0.00|0.00|0.61|0.60|3.48|3.48|1.76|1.01|1.00|1.00|
> > |**Fine-tuned Model Performance**|1.00|1.00|1.00|1.00|1.85|0.55|1.00|1.00|0.00|0.01|0.56|0.58|1.55|1.58|1.03|0.53|0.49|0.61|
> > |**Random Chance**|0.50|0.50|0.50|0.50|-|-|0.50|0.25|-|-|0.167|0.50|-|-|-|-|-|-|
> > |**Human Performance**|0.97|0.98|0.98|0.99|1.73|0.30|1.00|1.00|0.00|0.00|0.84|0.54|2.73|-|2.04|1.23|1.20|1.22|
> >
> > ## **Insights**
> >
> > * **Finetuning improves performance but does not solve thermal reasoning.**
> > SFT reduces MAE and improves accuracy, indicating that VLMs have the latent capacity to handle thermal data. However, the remaining errors are still substantial: absolute temperature estimates deviate by 1–2 °C in T6/T7, and performance on semantic comparison tasks (T5) remains below human-level. For applications such as fever screening or industrial hotspot detection, such error margins are not acceptable and highlight the need for deeper physical grounding. For instance, a standard deviation of 1-2 °C would be unsuitable for any model intended for non-contact fever detection.
> > * **SFT closes the gap because current VLMs lack domain grounding, not capacity.**
> > VLMs do not inherently understand temperature as a physical quantity or thermal appearance as a modality, even though they can acquire these concepts with minimal supervision. The fact that small-scale finetuning resolves failures on basic tasks suggests that the primary limitation lies in incomplete training signals rather than in model architecture or scale.
> > * **ThermEval isolates primitive abilities that RGB-centric pretraining does not teach.**
> > Pretrained VLMs, which are predominantly exposed to RGB photographs, diagrams, and charts, tend to learn mappings from appearance to semantic categories. Thermal understanding, however, requires mapping appearance to a physical quantity such as temperature. Because current models are not trained on this type of signal, they do not naturally acquire it, and SFT can only partially bridge this gap.
> > * **Future VLM pretraining should include physical sensor modalities.**
> > Most existing VLMs are trained primarily on RGB imagery, and although the training data of closed-source models are not public, available evidence and model behavior suggest limited exposure to thermal infrared data. This likely contributes to why current models interpret thermal images as RGB-like visuals rather than as physical measurements. Recent efforts, such as the Gemini team’s inclusion of modalities like X-rays and CT scans, and similar advances in remote-sensing VLMs, demonstrate that expanding pretraining beyond RGB is both feasible and beneficial. These developments indicate that incorporating additional physical sensing modalities is an important direction for future VLM development.

---

> > > ### Author Response · Authors · 2025-11-25
> > > **A Benchmark of all False Colormap Representations**
> > >
> > > Reiterating the reviewer concern:
> > >
> > > **R#c8GH**
> > >
> > > (3) *"all kinds of fake color images ... have the meaning of being made into benchmarks. I look forward to including such content..."*
> > >
> > > ---
> > >
> > > ### **A Benchmark of Colormaps**
> > > We appreciate the reviewer’s perspective and agree that a unified investigation of pseudo color representations such as PCA colored hyperspectral data, attention heatmaps, segmentation visualizations, and thermal imagery is an interesting and meaningful direction. This could indeed form a broader and more comprehensive benchmark. However, incorporating all such representations would significantly expand the scope of the current work, which is focused specifically on evaluating VLMs on thermal imagery. We believe this narrower scope is appropriate for the present paper, but we also agree that the reviewer’s suggestion raises an important point for the community. To acknowledge this and motivate future work, we are happy to add a discussion section highlighting the potential for a broader benchmark on pseudo color imagery.

---

> > > > ### Author Response · Authors · 2025-11-25
> > > > **Prompts Ablation Experiment**
> > > >
> > > > In this message, we address the prompt-ablation experiments requested by R#Ath4. We conducted a systematic study across InternVL, MiniCPM, Qwen-VL-2.5, and BLIP2, comparing the original zero-shot prompts with context-augmented versions.
> > > >
> > > > ---
> > > >
> > > > ## **Prompts Ablation**
> > > >
> > > > We first present the ablation examples below. In these experiments, we added contextual information about the thermal images to guide the VLM toward the relevant aspects of the scene.
> > > >
> > > > ### **Task T1,T2**
> > > >
> > > > * **Original:** "Is this a thermal image or an RGB image?"
> > > > * **Ablation:** "RGB images come from visible light and depict natural color and texture. Each pixel represents the intensity of red, green and blue channels that together form the visual appearance of objects under illumination.
> > > > Thermal images measure emitted infrared radiation and encode temperature dependent signals. Each pixel represents a temperature value or a value proportional to heat emission, and any colors seen in the image come from a colormap applied to these underlying temperature readings
> > > > Based on above context is the given image an RGB or Thermal Image?"
> > > >
> > > > ### **Task T6**
> > > > * **Original:** "Given the thermal image, what is the temperature at the coordinates ({x},{y})? The temperature scale is in degrees Celsius. Return a single numerical value rounded to one decimal place (e.g., 17.6)."
> > > > * **Ablation:** "Thermal images encode temperature dependent infrared signals. Each pixel represents a temperature value or a value proportional to emitted heat, and any visible colors come from a colormap applied to these values. A visible color bar or temperature scale maps colormap colors to numeric temperature readings in degrees Celsius. Given image pixel coordinates ({x},{y}) with origin at the top-left, x increasing to the right and y increasing downward, report the temperature at the specified coordinates as a single numeric value in degrees Celsius rounded to one decimal place (for example, 17.6)."
> > > >
> > > > We have not shown all examples here due to limited space, but we performed similar prompt ablations for all other tasks and subtasks in the same spirit.
> > > >
> > > > ---
> > > >
> > > > ### **Table demonstrates the performance of models before and after ablations. ↓ means lower is better, and ↑ means higher is better**
> > > > |Tasks|Dataset|InternVL-14B (Original)|InternVL-14B (Ablation)|MiniCPM (Original)|MiniCPM (Ablation)| Qwen_2.5_VL-7B (Original)|Qwen_2.5_VL-7B (Ablation)| Blip2 (Original)|Blip2 (Ablation)|
> > > > |-|-|-|-|-|-|-|-|-|-|
> > > > |T1|FLIR ↑|0.96|0.97|0.95|0.96|0.71|0.96|0.46|0.34|
> > > > ||LLVIP ↑|1.00|1.00|0.98|0.99|0.72|0.96|0.22|0.43|
> > > > |T2|FLIR ↑|0.86|0.99|0.91|0.89|0.61|0.98|0.77|0.67|
> > > > ||LLVIP ↑|0.97|1.00|0.93|0.95|0.80|0.99|0.77|0.77|
> > > > |T3|FLIR ↓|2.70|2.53|3.70|2.90|3.78|3.20|4.69|4.65|
> > > > ||LLVIP ↓|0.73|0.60|1.09|0.80|1.09|0.88|2.99|2.94|
> > > > |T4|Detection ↑|1.00|1.00|1.00|1.00|1.00|1.00|0.50|0.50|
> > > > ||Position ↑|1.00|1.00|0.99|0.99|0.99|0.97|0.25|0.25|
> > > > ||Min ↓|0.00|0.00|0.00|0.00|2.66|1.36|42.58|25.62|
> > > > ||Max ↓|0.00|0.00|0.00|0.00|0.00|0.00|209.80|20.80|
> > > > |T5|Single ↑|0.32|0.20|0.27|0.26|0.42|0.39|0.16|0.42|
> > > > ||Double ↑|0.51|0.50|0.40|0.41|0.41|0.51|0.39|0.39|
> > > > |T6|Arrow ↓|5.28|4.30|6.32|5.95|4.75|4.46|12.74|12.77|
> > > > ||Coordinate ↓|3.48|4.12|4.00|4.85|3.65|4.22|13.08|12.74|
> > > > ||Region ↓|2.18|2.51|4.23|3.29|2.91|3.22|14.73|14.73|
> > > > |T7|2ft ↓|1.01|2.04|2.15|2.45|1.05| 1.63|16.96|16.96|
> > > > ||6ft ↓|1.12|2.26|2.02|2.47|1.00|1.28|16.35|16.35|
> > > > ||10ft ↓|1.70|2.53|1.85|2.01|1.00|1.23|15.43|15.43|
> > > >
> > > > ---
> > > >
> > > >
> > > > ## **Key Findings**
> > > > Our ablation reveals three clear trends across models and tasks.
> > > > * Models with reasonable visual grounding (InternVL-14B, MiniCPM, Qwen-VL-2.5) show large gains for simple tasks when contextual modality descriptions are added.
> > > >     - Qwen-VL-2.5 — T1 FLIR: 0.71 → 0.96 and LLVIP: 0.72 → 0.96
> > > >     - Qwen-VL-2.5 — T2 FLIR: 0.61 → 0.98 and LLVIP: 0.80 → 0.99
> > > >     - InternVL-14B — T2 FLIR: 0.86 → 0.99 and LLVIP: 0.97 → 1.00
> > > >
> > > >     **Interpretation:** Architecture is not the Bottleneck; a short textual description helps models *anchor* the visual signal and correctly name the modality.
> > > > * Across temperature-comparison and temperature-estimation tasks gains are small, inconsistent, or negative.
> > > >     - Qwen T6 Arrow: 4.75 → 4.46 (small improvement)
> > > >     - MiniCPM T6 Coordinate: 4.00 → 4.85 (worse)
> > > >     - InternVL T7 (2ft): 1.01 → 2.04 (worse)
> > > >
> > > >     **Interpretation:** Thermal-physics descriptions in prompts cannot replace the sensor-level priors needed to map pixel or colormap values to temperature. The underlying challenge is the lack of thermal-domain grounding in model pretraining.
> > > >
> > > > * BLIP-2 often degrades when given extra context:
> > > >     - T1 FLIR: 0.46 → 0.34
> > > >     - T2 FLIR: 0.77 → 0.67
> > > >     - T6/T7: unchanged and poor.
> > > >
> > > >     **Interpretation:** When a model lacks basic thermal–visual alignment, prompt engineering cannot compensate for that gap. This aligns with the reviewer’s intuition that prompting alone is insufficient for reliable thermal reasoning.
> > > >
> > > > ---
> > > >
> > > > We will include these findings in the final manuscript.

---

> ### Author Response · Authors · 2025-11-29
> **Improved LLM-as-a-judge pipeline**
>
> ## **Improved LLM-as-a-judge pipeline**
>
> We thank the reviewer Ath4 for suggesting the use of structured outputs for evaluation. To provide additional context, below quote is taken from the original review of R#Ath4.
>
> >The authors decided to use LLM-as-a-judge because the outputs often vary structurally (L314), resulting in 97% consistent compared to the human evaluation (L348). However, multiple libraries, such as VLLM, have support structure output for open-source VLMs and the authors can try to use that to make it better when it's a easy regression or yes/no question.
>
> We explored this direction using vLLM and related libraries (Instructor, Outlines,etc), but found that they do not support all VLMs included in our benchmark. This prevents us from applying a uniform evaluation procedure to every model.
>
> Instead, we enforce structure within our LLM as judge framework, which allows us to parse and score answers reliably regardless of the underlying VLM. **Following the reviewer’s suggestion, we additionally experimented with structured judging using Gemini 2.5 Pro, Gemini 2.5 Flash, and Gemini 2.5 Flash Lite together with the Instructor library. This improves judge agreement from 97 percent to 99.01 percent, 99.07 percent, and 98.24 percent respectively.**
>
> We thank the reviewer again for this helpful suggestion. It led us to strengthen our evaluation pipeline and report a more robust judge-agreement analysis. We will include these findings in the camera ready version of the manuscript as well.

---

### Official Review · Reviewer_c8GH · 2025-10-31

**Soundness:** 3
**Presentation:** 3
**Contribution:** 3
**Rating:** 6
**Confidence:** 3

**Summary:**

The paper proposed a benchmark to evaluate the general-purpose vision-language models (VLMs) on thermal images. The benchmark includes:
- ThermEval-D: the first dataset with per-pixel temperature annotations across diverse environments.
- ThermEval-B: A comprehensive visual question-answering benchmark.
Besides, they conduct various evaluations for latest VLMs.

**Strengths:**

- A novel idea that evaluate on thermal images.
- The dataset with pixel-by-pixel temperature annotation (ThermEval-D) is unique in existing literature.
- The design comprises seven levels, ranging from simple to complex.
- A comprehensive and detailed evaluation and analysis.
- Excellent reproducibility

**Weaknesses:**

- The authors were expected to conduct further analysis on some failure cases.
- The evaluated models are mostly open-sourced model with small sizes. What about the closed-source product like Seed-VL and GPT-5? and large open-source models (like Qwen3VL-A22B)

**Questions:**

I agree with trying zero-shot testing on these types of images, but I think the gap between our model's vision and the human eye is huge. Heatmaps seem to be created by the human eye, and if we disregard accurate interpretation, they are naturally perceptible to humans. However, the model's vision is data-driven, and to my knowledge, our data is severely lacking in such thermal imaging samples. Therefore, it seems inevitable that future research will involve data in this area, perhaps even domain-specific VLM. I'd like to ask the authors how they view this.

---

> ### Author Response · Authors · 2025-11-19
> **Evaluating proprietary models, larger open-source models and fine-tuning.**
>
> We thank the reviewers for their constructive feedback. All three reviewers requested evaluations with: i) proprietary, ii) larger open-source, and iii) finetuned models. We have incorporated these additions and report the results below.
>
> ## **Proprietary and Large Open Source Models**
>
> We have extended our analysis to include some proprietary and large open source models,
> * GPT 4o,
> * Gemini 2.5 Flash,
> * Claude Haiku,
> * Qwen A22, a large open source model with 235B parameters.
>
> These models were evaluated on Tasks T5 (thermal reasoning), T6 (temperature estimation), and T7 (temperature estimation at varying depth). We focused on these tasks because they are the most challenging. Some open source models already perform near perfectly on the remaining tasks, leaving little room for meaningful comparison.
>
> We report the result in the table below.
>
> ### **Comparison of VLM performance on Tasks 5, 6, and 7. x means the model did not answer, ? means the model size is unknown, ↓ means lower is better, and ↑ means higher is better.**
> |Model|Params|T5 Double (Acc ↑)|T5 Single (Acc ↑)|T6 Coords (MAE ↓)|T6 Arrow (MAE ↓)|T6 Region (MAE ↓)|T7 2ft (MAE ↓)|T7 6ft (MAE ↓)|T7 10ft (MAE ↓)|
> |-|-|-|-|-|-|-|--|-|-|
> |**Qwen A22**|235B|0.58|0.27|3.96|4.21|3.01|1.97|2.09|2.24|
> |**Gemini 2.5 Flash**|?|0.54|0.28|3.81|**3.48**|2.50|1.30|1.80|1.96|
> |**GPT 4o**|?|0.46|0.34|x|x|x|x|x|x|
> |**Claude Haiku 4.5**|?|0.28|**0.60**|4.28|4.45|2.47|1.37|1.57|1.90|
> |**Best open source Performance**|max 38B|**0.61**|0.42|**3.48**|4.21|**1.76**|**1.01**|**1.00**|**1.00**|
> |**Human Performance**|--|0.84|0.54|--|2.73|2.04|1.23|1.20|1.22|
>
>
> ### **Findings**
> * Larger open source and proprietary models still struggle with thermal reasoning and the performance gap between humans and VLMs persists.
> * Increasing model size does not improve performance on thermal tasks, and larger models (Qwen A22 235B) show the same qualitative limitations as smaller ones (max 38B).
> * GPT 4o declines to provide temperature estimates and returns responses such as “I cannot determine the exact temperature” or “I am unable to provide an exact temperature estimate”. We view this as an appropriate safety guardrail because a reliable VLM should avoid confident answers when uncertain.
> * On MCQ based (Task-5 ordering body parts) tasks where GPT 4o does respond, its performance is comparable to other open source models and remains below human accuracy.
>
> ---
>
> ## **Finetuning Experiments**
> We fine tuned Qwen 2.5 VL with 7B parameters on the ThermEval dataset using LoRA adapters to illustrate the dataset’s utility. The setup and results are summarised below.
>
> ### **Comparison of finetuned Qwen VL 2.5 (7B) with human performance and all other models. ↓ means lower is better, ↑ means higher is better, and - indicates not applicable**.
> |Task|Best Model (Zeroshot)|Human Performance|Qwen 2.5 7B Zeroshot|Qwen 2.5 7B Finetuned| Qwen Δ (Finetuned − Zeroshot)|
> |-|-|-|-|-|-|
> |**T1 FLIR (Acc ↑)**|**1.00**|0.97|0.71|**1.00**|+0.29|
> |**T1 LLVIP (Acc ↑)**|**1.00**|0.98|0.71|**1.00**|+0.29|
> |**T2 FLIR (Acc ↑)**|**1.00**|0.98|0.61|**1.00**|+0.39|
> |**T2 LLVIP (Acc ↑)**|**1.00**|0.98|0.80|**1.00**|+0.20|
> |**T3 FLIR (MAE ↓)**|2.72|**1.73**|3.78|1.85|−1.93|
> |**T3 LLVIP (MAE ↓)**|0.51|**0.30**|1.09|0.55|−0.54|
> |**T4 Detect (Acc ↑)**|1.00|**1.00**|1.00|**1.00**|0.00|
> |**T4 Position (Acc ↑)**|1.00|**1.00**|0.99|**1.00**|+0.01|
> |**T4 Max (MAE ↓)**|0.00|**0.00**|0.00|**0.00**|0.00|
> |**T4 Min (MAE ↓)**|0.00|**0.00**|2.66|**0.00**|−2.66|
> |**T5 Double (Acc ↑)**|0.61|**0.84**|0.41|0.58|+0.17|
> |**T5 Single (Acc ↑)**|**0.60**|0.54|0.42|0.56|+0.14|
> |**T6 Cords (MAE ↓)**|3.48|--|3.65|**1.58**|−2.07|
> |**T6 Arrow (MAE ↓)**|3.48|2.73|4.75|**1.55**|−3.20|
> |**T6 Region (MAE ↓)**|1.76|2.04|2.91|**1.03**|−1.88|
> |**T7 2ft (MAE ↓)**|1.01|1.23|1.05|**0.53**|−0.52|
> |**T7 6ft (MAE ↓)**|1.00|1.20|1.00|**0.49**|−0.51|
> |**T7 10ft (MAE ↓)**|1.00|1.22|1.00|**0.61**|−0.39|
>
> ### **Findings**
> * Finetuning Qwen VL 2.5 allows it to outperform the much larger Qwen A22 235B and all evaluated closed source as well as open source models.
> * The finetuned model matches or exceeds human performance on most tasks.
> * These results show that ThermEval delivers meaningful supervision for thermal reasoning and establishes a reliable benchmark for advancing thermal modality capabilities in VLMs.
>
> ### **Experimental Setup**
> * We fine tuned Qwen2.5 VL 7B Instruct for 5 epochs using LoRA with rank (r = 16) and scaling factor (alpha = 16), applied to the query, key, value, and output projection layers with dropout (0.1). Training used Paged AdamW 32 bit with a fixed learning rate (5 x 10^-6), no warmup, batch size 4 per device.
> * The dataset was split into three stratified folds to balance tasks and subtasks, ensuring each VQA sample was seen exactly once and enabling full dataset evaluation without repetition.

---

> ### Comment · Reviewer_c8GH · 2025-11-23
>
> Thanks for the rapid response.
> I'm also discussing in Ath4's space.
>
> I am very grateful for the author's additional experiments.
>
> I think that the training data of the closed source model is uncertain, so it is not clear whether the evaluation is indeed zeroshot, but this is a relatively serious statement. I think the author has done enough.
>
> Because I don't know much about thermal imaging, I don't give much confidence. But I will actively participate in the discussion to treat the work fairly.
>
> I am more concerned about whether the author's benchmark can match and cover the application of thermal imaging, which is something that my professional background cannot be clear about. I care more about the recruitment benchmark that does solve the evaluation problem in this field, and it is difficult to be surpassed. If the authors have more confidence about this, please provide more details.

---

### Official Review · Reviewer_f9Nx · 2025-10-31

**Soundness:** 3
**Presentation:** 3
**Contribution:** 2
**Rating:** 6
**Confidence:** 3

**Summary:**

This paper presents ThermEval, a comprehensive benchmark and dataset for evaluating VLMs on thermal imagery. It includes ThermEval-B, a 50k-sample benchmark across seven thermal reasoning tasks, and ThermEval-D, a 500-image dataset with per-pixel temperature annotations and body-part segmentation. The authors benchmark 14 open-source VLMs (e.g., LLaVA, Intern-VL, Qwen-VL) under strict zero-shot settings and reveal systematic weaknesses in thermal reasoning. Current models rely heavily on language priors, misread colorbars, and fail at temperature estimation even when they handle RGB tasks well.

**Strengths:**

1. Addresses a under-explored but important domain for vision understanding, thermal imagery, in the context of multimodal reasoning.
2. Well-structured benchmark with increasing task difficulty, covering both perception (modality, counting) and reasoning tasks (temperature estimation).
3. Comprehensive experimental design: large-scale comparison of 14 VLMs under a unified zero-shot protocol.
4. Clear failure analysis with concrete qualitative examples (e.g., fixed numeric biases, hallucination from priors).
5. Ethical considerations and dataset release add reproducibility and impact.

**Weaknesses:**

1. Evaluation is purely zero-shot; no fine-tuning or adaptation experiments on existing VLMs to show the dataset potential.
2. The new dataset (ThermEval-D) is relatively small and limited to controlled environments, diversity and realism remain low.
3. It heavily depends on an LLM-as-a-judge introduces parsing noise and interpretability issues.
4. The authors didn't include proprietary or stronger close-sourced models like GPT-5/Gemini2.5, which limits the completeness.

**Questions:**

This paper shows a meaningful limitation of current VLMs and contributes a benchmark that will likely inspire follow-up work in thermal-aware multimodal learning.

Despite limited dataset scope and missing finetuning studies, I tend to accept this paper given its contribution.

**Details Of Ethics Concerns:**

The paper involves direct data collection from human participants for the ThermEval-D dataset, including per-pixel thermal imagery of identifiable body parts (forehead, chest, nose). The authors state that an Institutional Ethics Committee approved the study and that all data were anonymized.

Given these factors, an ethics review may be necessary that focused on human data collection, privacy safeguards, and responsible dataset release.

---

> ### Author Response · Authors · 2025-11-19
> **Evaluating proprietary models, larger open-source models and fine-tuning.**
>
> We thank the reviewers for their constructive feedback. All three reviewers requested evaluations with: i) proprietary, ii) larger open-source, and iii) finetuned models. We have incorporated these additions and report the results below.
>
> ## **Proprietary and Large Open Source Models**
>
> We have extended our analysis to include some proprietary and large open source models,
> * GPT 4o,
> * Gemini 2.5 Flash,
> * Claude Haiku,
> * Qwen A22, a large open source model with 235B parameters.
>
> These models were evaluated on Tasks T5 (thermal reasoning), T6 (temperature estimation), and T7 (temperature estimation at varying depth). We focused on these tasks because they are the most challenging. Some open source models already perform near perfectly on the remaining tasks, leaving little room for meaningful comparison.
>
> We report the result in the table below.
>
> ### **Comparison of VLM performance on Tasks 5, 6, and 7. x means the model did not answer, ? means the model size is unknown, ↓ means lower is better, and ↑ means higher is better.**
> |Model|Params|T5 Double (Acc ↑)|T5 Single (Acc ↑)|T6 Coords (MAE ↓)|T6 Arrow (MAE ↓)|T6 Region (MAE ↓)|T7 2ft (MAE ↓)|T7 6ft (MAE ↓)|T7 10ft (MAE ↓)|
> |-|-|-|-|-|-|-|--|-|-|
> |**Qwen A22**|235B|0.58|0.27|3.96|4.21|3.01|1.97|2.09|2.24|
> |**Gemini 2.5 Flash**|?|0.54|0.28|3.81|**3.48**|2.50|1.30|1.80|1.96|
> |**GPT 4o**|?|0.46|0.34|x|x|x|x|x|x|
> |**Claude Haiku 4.5**|?|0.28|**0.60**|4.28|4.45|2.47|1.37|1.57|1.90|
> |**Best open source Performance**|max 38B|**0.61**|0.42|**3.48**|4.21|**1.76**|**1.01**|**1.00**|**1.00**|
> |**Human Performance**|--|0.84|0.54|--|2.73|2.04|1.23|1.20|1.22|
>
>
> ### **Findings**
> * Larger open source and proprietary models still struggle with thermal reasoning and the performance gap between humans and VLMs persists.
> * Increasing model size does not improve performance on thermal tasks, and larger models (Qwen A22 235B) show the same qualitative limitations as smaller ones (max 38B).
> * GPT 4o declines to provide temperature estimates and returns responses such as “I cannot determine the exact temperature” or “I am unable to provide an exact temperature estimate”. We view this as an appropriate safety guardrail because a reliable VLM should avoid confident answers when uncertain.
> * On MCQ based (Task-5 ordering body parts) tasks where GPT 4o does respond, its performance is comparable to other open source models and remains below human accuracy.
>
> ---
>
> ## **Finetuning Experiments**
> We fine tuned Qwen 2.5 VL with 7B parameters on the ThermEval dataset using LoRA adapters to illustrate the dataset’s utility. The setup and results are summarised below.
>
> ### **Comparison of finetuned Qwen VL 2.5 (7B) with human performance and all other models. ↓ means lower is better, ↑ means higher is better, and - indicates not applicable**.
> |Task|Best Model (Zeroshot)|Human Performance|Qwen 2.5 7B Zeroshot|Qwen 2.5 7B Finetuned| Qwen Δ (Finetuned − Zeroshot)|
> |-|-|-|-|-|-|
> |**T1 FLIR (Acc ↑)**|**1.00**|0.97|0.71|**1.00**|+0.29|
> |**T1 LLVIP (Acc ↑)**|**1.00**|0.98|0.71|**1.00**|+0.29|
> |**T2 FLIR (Acc ↑)**|**1.00**|0.98|0.61|**1.00**|+0.39|
> |**T2 LLVIP (Acc ↑)**|**1.00**|0.98|0.80|**1.00**|+0.20|
> |**T3 FLIR (MAE ↓)**|2.72|**1.73**|3.78|1.85|−1.93|
> |**T3 LLVIP (MAE ↓)**|0.51|**0.30**|1.09|0.55|−0.54|
> |**T4 Detect (Acc ↑)**|1.00|**1.00**|1.00|**1.00**|0.00|
> |**T4 Position (Acc ↑)**|1.00|**1.00**|0.99|**1.00**|+0.01|
> |**T4 Max (MAE ↓)**|0.00|**0.00**|0.00|**0.00**|0.00|
> |**T4 Min (MAE ↓)**|0.00|**0.00**|2.66|**0.00**|−2.66|
> |**T5 Double (Acc ↑)**|0.61|**0.84**|0.41|0.58|+0.17|
> |**T5 Single (Acc ↑)**|**0.60**|0.54|0.42|0.56|+0.14|
> |**T6 Cords (MAE ↓)**|3.48|--|3.65|**1.58**|−2.07|
> |**T6 Arrow (MAE ↓)**|3.48|2.73|4.75|**1.55**|−3.20|
> |**T6 Region (MAE ↓)**|1.76|2.04|2.91|**1.03**|−1.88|
> |**T7 2ft (MAE ↓)**|1.01|1.23|1.05|**0.53**|−0.52|
> |**T7 6ft (MAE ↓)**|1.00|1.20|1.00|**0.49**|−0.51|
> |**T7 10ft (MAE ↓)**|1.00|1.22|1.00|**0.61**|−0.39|
>
> ### **Findings**
> * Finetuning Qwen VL 2.5 allows it to outperform the much larger Qwen A22 235B and all evaluated closed source as well as open source models.
> * The finetuned model matches or exceeds human performance on most tasks.
> * These results show that ThermEval delivers meaningful supervision for thermal reasoning and establishes a reliable benchmark for advancing thermal modality capabilities in VLMs.
>
> ### **Experimental Setup**
> * We fine tuned Qwen2.5 VL 7B Instruct for 5 epochs using LoRA with rank (r = 16) and scaling factor (alpha = 16), applied to the query, key, value, and output projection layers with dropout (0.1). Training used Paged AdamW 32 bit with a fixed learning rate (5 x 10^-6), no warmup, batch size 4 per device.
> * The dataset was split into three stratified folds to balance tasks and subtasks, ensuring each VQA sample was seen exactly once and enabling full dataset evaluation without repetition.

---

> > ### Author Response · Authors · 2025-11-25
> > **A gentle reminder**
> >
> > Hello Reviewer f9Nx, We wanted to check if you had a chance to look at our response to your questions.
> >
> > The other two reviewers posted more follow up questions, and we have added our responses to them as well.
> >
> > We request you to please consider those as well.

---

> > ### Comment · Reviewer_f9Nx · 2025-11-27
> >
> > Thanks for the additional experiments. That solve part of my concerns.
> >
> > However, the scope also start to feels a bit narrow to me. The authors introduce a dataset of a new thermal modality that current VLMs have never seen and state that this modality matters, and with the additional data to finetune VLMs, the performance can improve. This pattern could apply to almost any new modality, and the experiment outcome feels somewhat expected.
> >
> > I think the key question is if it's even a good way to let a VLM analyze thermal data. Unlike RGB images, a thermal image is for human visualization. A machine can read the raw radiance values directly without looking at the image at all. In that sense, it does not make much sense to finetune a VLM to work on a modality that exists only for human viewing. A specialized model that takes the raw sensor input would be a more suitable choice.
> >
> > I'm curious what other reviewers think about this and I'd love to join the discussion.

---

> > > ### Author Response · Authors · 2025-11-29
> > > **False Colormapped Thermal Imagery Is the Standard Interface for Thermal Modality**
> > >
> > > Thank you for raising these points. We summarize and respond to your core concerns here and in the following messages.
> > >
> > > ---
> > > ## **Visualized Physical Modalities Are Standard and Effective for VLM Reasoning**
> > >
> > > > “The scope feels narrow..current VLMs have never seen this modality, so the results feel expected.”
> > >
> > > > “Is it even sensible to let a VLM analyze thermal data? Machines could read raw radiance directly rather than looking at the rendered image.”
> > >
> > > > “Should we instead visualize every modality (depth, normals, segmentation) and let VLMs reason?”
> > >
> > > We appreciate the reviewer’s concern. Our approach follows a standard practice in VLM research, where physical measurements are converted into visually interpretable representations. Many recent works show that VLMs learn reliably from false colormapped physical modalities and that this is the usual operable form for depth, thermal, and other sensor data. **VLMs are not limited to RGB, only predominantly trained on it.**
> > >
> > > * **SpatialBot (ICRA 2025)**[1]: Uses depth rendered as false colormaps for spatial reasoning, motivated by the same observation that VLMs excel with RGB visual inputs but need visualized physical cues for modality specific reasoning.
> > > * **Leveraging MLLMs for Thermal Perception in Autonomous Driving (MDPI 2024)**[2]: Shows that GPT and Gemini can interpret thermal images directly, confirming that thermal colormaps support high level understanding tasks.
> > > * **IRGPT (ICCV 2025)**[3]: Uses false-color thermal images primarily for detection and scene interpretation, but does not address semantic to physical reasoning (temperature estimation, ranking, or interpreting colorbars). ThermEval therefore targets a different capability class not covered in prior work.
> > > * **Thermal Image Driven Clothing Insulation Estimation (Energy and Buildings 2025)** [4]: Uses false colormapped thermal images for quantitative prediction of clothing insulation, reinforcing that thermal visualizations are a standard modality for extracting physically meaningful quantities.
> > > * **Zero Shot Anomaly Detection in Battery Thermal Images (EUSIPCO 2025)** [5]: Applies pretrained VQA models to false colormapped thermal images for battery anomaly detection, showing that text guided reasoning over thermal colormaps can support diagnostic tasks without modality specific training.
> > >
> > > Together, these works show that false color thermal images are the standard operable form of the modality and are widely used across detection, reasoning, and prediction tasks in foundational and applied settings.
> > >
> > > Below are representative works where models predict physical quantities from visual inputs, aligning with our temperature estimation task
> > > * DepthLM (Meta) [6]: Predicts metric depth from RGB with light supervised finetuning and reaches accuracy comparable to specialized depth models, showing that VLMs achieve expert level performance without architectural or loss changes.
> > > * Depth and Height Perception in Large VLMs (CVPRW 2025) [7]: Demonstrates that VLMs infer 3D structure and height from visual encodings rather than raw physical measurements, emphasizing that visualization is the mechanism enabling geometric reasoning.
> > >
> > > These efforts show that predicting physical quantities from image inputs is well established and effective. They support three conclusions
> > > 1. VLMs operate on visual representations, not raw signals.
> > > 2. Visualized physical modalities are valid and widely used inputs.
> > > 3. Small supervised finetuning is sufficient to unlock physical grounding.
> > >
> > > Our paper follows the same principle. Once thermal signals are rendered visually, they form a modality that VLMs can naturally learn from, mirroring patterns seen in depth and other physical signals.
> > >
> > > Regarding the question,
> > > >“If VLMs need SFT to handle this, is this approach meaningful?”
> > >
> > > DepthLM [6] states: “SFT with sparse labels is sufficient to unlock strong 3D understanding, no architectural change is needed.” Our findings align with this line of work. VLMs already have the representational capacity, and the main missing component is exposure to the modality. We show that thermal signals, when visualized in a principled manner, follow the same learning pattern observed for depth and other physical modalities.
> > >
> > > ---
> > > [1] SpatialBot: Precise Spatial Understanding with VLMs , ICRA'25 https://ieeexplore.ieee.org/document/11128671
> > >
> > > [2] Leveraging MLLMs for Thermal Perception in Autonomous Driving, MDPI'24  https://www.mdpi.com/2673-4052/5/4/29
> > >
> > > [3] IRGPT:  ICCV'25 https://arxiv.org/abs/2507.14449
> > >
> > >
> > > [4] Thermal Image Driven Clothing Insulation Estimation, Energy and Buildings 2025. https://doi.org/10.1016/j.enbuild.2025.116720
> > >
> > > [5] Zero Shot Anomaly Detection in Battery using Thermal Images, EUSIPCO'25 https://eusipco2025.org/wp-content/uploads/pdfs/0000711.pdf
> > >
> > > [6] DepthLM : Metric Depth From Vision-Language Models” https://arxiv.org/pdf/2509.25413
> > >
> > > [7] Understanding Perception in Large Visual-Language.  CVPR'25 https://arxiv.org/abs/2408.11748

---

### Author Response · Authors · 2025-12-03
**Author Summary [1/2]**

We thank the ACs for overseeing the review process. We have addressed all requested changes and incorporated the corresponding updates into the revised manuscript. Below, we summarize the major additions and improvements.

---

# Paper Summary
This paper presents the first systematic evaluation of VLMs on thermal imagery, an important yet under explored visual modality with applications in autonomous driving, medical screening, and industrial monitoring. We introduce a comprehensive benchmark and analyze how current VLMs perform on both thermal perception and reasoning.

**Key contributions:**
* 50,000 VQA across seven tasks spanning perception to advanced reasoning, including a pixel-level annotated Dataset.
* Evaluation of 21 VLMs, enabling the first large-scale study on thermal modality.
* Analysis of key failure modes (numeric bias, prior-driven hallucinations, weak physical grounding).
* A fully reproducible evaluation pipeline with transparent protocols, ethical considerations, and a publicly released dataset.
* A foundation for future work on multimodal thermal understanding and reasoning.

---
# Strengths as per reviewers

**R#f9Nx**
* Addresses a under-explored but important domain for vision understanding, thermal imagery, in the context of multimodal reasoning.
* Well-structured benchmark with increasing task difficulty, covering both perception (modality, counting) and reasoning tasks (temperature estimation).
* Comprehensive experimental design: large-scale comparison of 14 VLMs under a unified zero-shot protocol.
* Clear failure analysis with concrete qualitative examples (e.g., fixed numeric biases, hallucination from priors).
* Ethical considerations and dataset release add reproducibility and impact.

**R#c8GH**
* A novel idea that evaluate on thermal images.
* The dataset with pixel-by-pixel temperature annotation (ThermEval-D) is unique in existing literature.
* A comprehensive and detailed evaluation and analysis.
* Excellent reproducibility

**R#Ath4**
* The authors present a new thermal evaluation dataset with precise annotation.
* The paper provides an initial study on zero-shot VLM capabilities in the thermal image domain.
* I appreciate the paper's contribution of releasing a large-scale, precise thermal image dataset, which can be good resources for future model training.

---

# Proprietary and large open source models.

All three reviewers requested evaluations with closed-source and larger open-source models. In response, we added
* Gemini-2.5,
* GPT-4o,
* Claude Haiku,
* Qwen3 A22 (235B)

The results consistently show that current VLMs remain insufficiently trained for reasoning over thermal imagery. The results are available in table-2 & 3 in our revised manuscript.

---
# Finetuning experiment
Reviewers requested we showcase the effect of finetuning. We show that finetuning Qwen-VL 2.5 (7B) matches or exceeds human performance on most tasks. Results are in Tables 2 and 3, with full details in Appendix B.6.

Questions were also raised about whether SFT reduces the insight of our study. Our findings highlight two points:
* **Finetuning improves performance but remains insufficient.** MAE drops, yet temperature estimation and semantic comparison errors are still too high for practical use.
* **Remaining failures reflect missing thermal grounding, not capacity.** Small scale SFT fixes basic issues, indicating that models mainly lack exposure to thermal concepts.

Finetuning narrows the gap but does not solve the underlying problem

---
# Improving the Judge pipeline
Two reviewers raised concerns about using an LLM as judge and recommended libraries with structured outputs. We explored vLLM, Outline, Instructor, etc but several models in our benchmark were unsupported, preventing a unified evaluation. Instead, we added structured constraints directly to our judge and, following the reviewers’ suggestion, tested structured output generation with Gemini 2.5 Pro, Flash, and Flash Lite via Instructor. This improved judge agreement from 97 percent to 99.01 percent, 99.07 percent, and 98.24 percent. These updates are reflected in Section 4.2 of the revised manuscript.

---
# Prompt Ablation
R#Ath4 suggested that adding context could improves performance. We therefore ran a systematic prompt-ablation study on InternVL, MiniCPM, Qwen-VL-2.5, and BLIP-2, comparing zero-shot prompts with context-augmented variants. Details are now included in Appendix B.7
* Model with weak visual grounding (e.g., BLIP-2) degrade with added context, showing that prompting cannot fix poor visual alignment.
* Temperature comparison and estimation tasks show minimal or negative gains, indicating that textual cues cannot replace missing physical grounding
* Models with stronger visual grounding improve on simple modality identification when given brief modality descriptions
* Textual cues alone are insufficient. VLMs require actual exposure to the underlying visual domain to handle this modality effectively

---

> ### Author Response · Authors · 2025-12-03
> **Author Summary [2/2]**
>
> Here we summarise the open-ended discussions that got interrupted due to Openreview's identity bug.
>
> ---
> # Thermal images are a distinct modality
> All three reviewers noted limited familiarity with the thermal domain and were unsure how thermal images differ from charts. Since ACs may have similar question, we offer brief clarifications:
> * Thermal images and charts may appear similar because both can use colormaps, but they represent different signals. Any single channel measurement can be color mapped, and this alone does not make it a chart. The colorbar is simply a scale, not structured semantics.
> * Charts and heatmaps are human designed with fixed layouts, axes, grids, and predictable structure. Thermal images have none of this and instead reflect raw sensor measurements shaped by geometry, emissivity, materials, distance, and occlusion.
> * Our evaluation of chart models (ChartGemma, ChartInstruct, TinyCharts) shows that they perform worse than general purpose VLMs, reinforcing that thermal understanding requires interpreting a physical scene rather than reading a structured diagram.
>
> These results appear in Tables 2 & 3 of revised manuscript.
>
> ### **Insights drawn from SAM3**
>
> Segment Anything Model 3 supports the view that thermal images function as a meaningful non RGB visual modality. The appendix notes
>
> >“SAM 3 struggles to generalize to fine-grained out-of-domain concepts, particularly in visual domains such as thermal imagery, in a zero-shot manner”
>
> SAM3 also evaluates on ODinW13, which includes ThermalDP, a thermal detection dataset. Its behavior closely matches our observations:
> * zero shot performance is poor but few shot finetuning improves it
> * the limitation arises from missing modality grounding, not model capacity
>
> These findings directly align with our results and reinforce that evaluating VLMs on visualized thermal data is consistent with contemporary practice.
>
> For these reasons, we argue that thermal images are not charts. They constitute are a visual modality.
>
> ---
> # False colormaps are the standard representation for thermal
> Reviewer R#f9Nx also asked whether false colormaps are a valid representation of thermal data. Our approach follows established practice in research, where physical measurements are converted into visually interpretable forms. Prior work like SpatialBot (ICRA 2025), IRGPT (ICCV 2025), etc shows that VLMs learn reliably from false colormapped modalities. This is standard for thermal, depth, and other single channel sensors. VLMs are predominantly trained on RGB but are not restricted to it.
>
> Raw radiance is rarely accessible, and major thermal datasets such as FLIR ADAS, LLVIP, KAIST, etc, release only false colored images. This makes false colored thermal imagery the practical modality for downstream analysis. We have added this clarification to the related works section.
>
> Our detailed responses to the reviewers include all relevant references.
>
> ---
> # Benchmark tasks represent real world applications of thermal Modality
> Reviewers sought clarification on how our task design aligns with real-world thermal applications. We emphasized real-world use cases where thermal imagery is essential:
> * **Tasks T1, T2, and T4** assess whether VLMs have basic grounding in the thermal modality. Models missing these fundamentals are unlikely to succeed in real applications
> * **Task T3** evaluates thermal detection, essential in low light or low visibility settings such as autonomous driving, surveillance, and occupancy monitoring
> * **Task T5** tests reasoning over relative temperature patterns, which remains informative even when absolute accuracy is limited. Relevant applications include fever screening, heat stress monitoring, and human robot interaction
> * **Tasks T6 and T7** measure numerical temperature estimation, needed for industrial inspection, building diagnostics, medical screening, and robotic operation near heat sources
>
> These tasks are representative of practical thermal applications.
>
> ---
> # Conclusion
> The revisions substantially strengthen the submission. We added all reviewer requested experiments, including finetuning, prompt ablations, and larger VLMs, improved the judge pipeline, clarified domain distinctions, and situated the benchmark within emerging work on physical modalities. Reviewers agreed that the topic is important, the dataset is unique, the evaluation is thorough, and reproducibility is excellent. With the expanded analyses, stronger baselines, and clearer motivation, the paper now offers a solid foundation for studying thermal reasoning in VLMs. Our work provides (i) a unique pixel temperature annotated dataset, (i) the most comprehensive evaluation of VLMs on this modality to date, and (iii) clear empirical insight into current model limitations and how small scale finetuning reduces them.
>
> ---
>
> We hope these updates demonstrate the value and timeliness of ThermEval and we appreciate the committee’s thoughtful consideration.

---

### Note · Program_Chairs · 2026-01-17
**Submission Desk Rejected by Program Chairs**

The following references in this submission do not refer to real documents and/or have major errors in bibliographic information:

 Yuxuan Zhang et al. Seed-bench: Benchmarking multimodal llms with generative comprehension. arXiv preprint arXiv:2307.14430, 2023.
Chunyuan Zheng et al. Mm-vet: Benchmarking multimodal large language models for robustness and generalization. arXiv preprint arXiv:2403.09334, 2024.